# A Graph-Based Network Analysis of Global Coffee Trade—The Impact of COVID-19 on Trade Relations in 2020

Zsuzsanna Bacsi [1,*] , Mária Fekete-Farkas [1] and Muhammad Imam Ma'ruf [1,2]

1   Institute of Agricultural and Food Economics, Hungarian University of Agriculture and Life Sciences, 2100 Godollo, Hungary
2   Development Economics Study Program, Economic Sciences Department, Faculty of Economics, Universitas Negeri Makassar (UNM), Makassar 90221, Indonesia
*   Correspondence: bacsi.zsuzsanna@uni-mate.hu

**Abstract:** International trade relations have been considerably affected by the coronavirus pandemic. Our analysis was aimed at identifying its effect on the global trade network of green coffee beans, comparing the COVID-year 2020 to the pre-COVID year 2018. The methodology applied was that of social network analysis using trade value data for the above two years. Our results show that between the pre-pandemic and the pandemic years, the role of some major actors considerably changed, and many trade relationships were disrupted. Overall trade value decreased, and the number of trade connections also changed—some countries gained, but more countries lost compared to their former positions. The network measures, i.e., degree distribution, betweenness, closeness and eigenvector centralities, modularity-based clustering and the minimum spanning tree, were suitable for quantifying these changes and identifying differences between affected countries. The changes found between the two years are assumed to be due to the effects of the pandemic, but further analysis is needed to reveal the actual mechanisms leading to these results.

**Keywords:** coffee; global trade; social network analysis; Gephi; centrality; minimal spanning tree

## 1. Introduction

The analysis of trade relationships has been the key issue in a very extensive range of literature.

Coffee is one of the most traded agricultural commodities in the world, with 12.5 million coffee farms situated geographically between the tropics, mainly in low- to lower-middle-income countries [1]. Smallholder farmers cultivate two types of coffee beans (robusta and arabica) and are responsible for the production of 70% of coffee beans globally [1,2]. Local farmers sell coffee beans to first- and second-level traders, and large green coffee traders and multinational firms operate in international markets, where most coffee is exported as "green" (not roasted). The downstream coffee value chain (from the roaster to consumption) is usually developed in consuming countries [2]. The coffee green bean gives only a small proportion of the value of the final product. The price of green coffee was rather low till the summer of 2021 ($2.71 per kg in April 2021) but showed a speedy increase from summer 2021 ($3.54 per kg in August 2021, 4.51 $/kg in January 2022, 4.21 $/kg in July and August 2022) [3], due to international trading on the stock exchange and to weather and climate conditions influencing yields and growing areas [1]. The coffee market is expanding. Global demand for coffee has increased by more than 60% since the 1990s, driving the expansion of production and exports, with more than 70% of the production of green coffee exported [1]. The international coffee market has become more complex, with some non-producing countries increasing exports and trade of processed coffee gaining ground [2]. Coffee export represents a rather high proportion of total agricultural exports of the main producer countries from 1% in Mexico to 47% in Ethiopia, representing a crucial share of their export revenues [4].

The evolution of the coffee trade network has been subject to many changes in the balance of power among actors. Until the Second World War, the world coffee market was centralised in Brazil. In 1962 the first international coffee agreement (ICA) was signed by most producing and consuming countries, and up to 1989, the target price for coffee was set by the ICA regulatory system, with export quotas allocated to each producer. The system, though not free of problems, was successful in raising and stabilising coffee prices [5,6]. The ICA ended in 1989, and the governance of the system shifted from producing and consuming countries, and the coffee marketing systems were liberalised [2,7]. In the 1980s, producers controlled almost 20%, and consuming countries received 55% of total trade income, but after the collapse of ICA, during 1989–1990 and 1994–1995, the share to producers dropped to 13% while that of consumers increased to 78%. In the 1990s and 2000s, the introduction of Structural Adjustment Programs (SAPs) changed the policy landscape of coffee production. The global market regime was liberalised, but price volatility increased. In 2000–2004 the price of coffee slumped, creating major social problems across coffee-producing regions of the world [2]. The coffee value chain changed drastically after the deregulation of the coffee trade at the end of the 20th century. Many developing countries depend on coffee exports for the livelihood of millions of small producers. Local governments, aware of this dependence on exports, have tried to control and regulate operations, but this control passed into the hands of the large multinationals after the liberalisation of coffee. In recent decades the market has become more concentrated, and only the largest coffee traders have survived [8].

The standard theory of international trade goes back to the years between 1776 and 1826, to the publications of Adam Smith's *The Wealth of Nations* [9] and David Ricardo's *Principles of Economics* [10], establishing a theory of free trade. Smith emphasised the division of labour as the basis for lower labour costs, ensuring effective competition across countries. While Smith established the idea of absolute advantage, Ricardo introduced the concept of comparative advantage to achieve production efficiency leading to international specialisation and mutually gainful trade across countries. A further key development in classical trade theory was the Heckscher–Ohlin theory [11], defining returns to the two factors of production (labour and capital) to be at levels proportional to their respective material contribution valued at market prices. International trade theory developed to incorporate factor-endowment of production factors, stating that the scarce factors are to lose under free trade and benefit from protection [12]. A major departure from old trade theories was made by introducing scale economies in production, including the impact of increasing returns to scale, imperfect markets and product differentiation. The new trade theory emanates from the new growth theory, which emphasises the determinants of technological progress as well as the externalities that the development and application of new knowledge confer as explicit variables that determine economic growth [11,13,14]. These implied that some market actors might be able to influence the market, leading to imperfect competition, such as monopolistic competition, oligopoly, or monopoly. Terms of trade were seen to be a powerful tool to demonstrate the inequities of trade for developing countries. Trade became a tool in the peripheralisation and the development of 'core' regions [13,14]. Much research has been conducted into the efficiency gains of free trade and eliminating trade barriers, although many indirect forms still prevail, as is seen during the multilateral discussions within the WTO framework [14].

The theoretical literature on trade and competitiveness emphasises the dynamic aspects of comparative advantage over time. The empirical literature on comparative advantage usually employs the concept of revealed comparative advantage developed by Balassa in 1965 [15,16]. Several studies imply that export competitiveness will increase with declining relative trade costs, which will contribute to a stronger comparative advantage. When the transportation costs are small, comparative advantage can then be of longer duration [15]. Higher trade costs decrease the probability of survival in comparative advantage, while the level of economic development, the size of the country, the agri-food export

diversification, and being a new EU member state increases it, which has implications for the EU-27 member states and agri-food policies [15].

Comparative advantage is usually demonstrated by the RCA (Revealed Competitive Advantage) index, which compares a country's export share of the product relative to its overall export share in global exports [16]. Analysis of bilateral trade has often been carried out by applying the gravity model. The model states that trade is enhanced by the economic size of the trading partners and hindered by the geographical distance between them, although other features of the trading partners, e.g., a common language or a common border, can be included in the model. Gravity equations have been extensively applied for the assessment of trade policy impacts. For references, see [17,18]. This tool was used together with RCA to test if there is a positive link between the size of trade flows and the extent to which they follow the pattern of comparative advantage, and it found that countries trading more with each other tend to follow the patterns of comparative advantages more than countries with smaller mutual trade flows [19].

An understanding of the operations of international trade networks is essential for achieving international food security. Trade networks connect countries around the world through the import-export flows of commodities. Import–export linkages can boost shock diffusion: higher connectivity in the network can lead to increasing fragility, as happened during the 2007–2008 global financial crisis. In recent years, much research focused on exploring the network architecture of aggregate and commodity-specific trade networks. Particular attention was paid to the community structure of food networks, i.e., clusters of actors characterised by a higher 'within-group' connectivity and much sparser connectivity between actors belonging to different clusters. Such clusters can be considered a proxy for geopolitical relations, which vary depending on the commodity and evolve over time. Identifying communities in a network is crucial for understanding its structure, its robustness, and shock transmissions [20]. Investments in transportation infrastructure and trade agreements have helped to increase the connectivity between nations over the last several decades to promote efficiency and enhance resilience. Higher connectivity may enable consumers to access a commodity from a variety of sources, but it may enable production shocks and export restrictions to be transmitted to importers [21].

Recently an increasing volume of research has dealt with the application of social network analysis for evaluating various aspects of the global or regional patterns of export and import relationships. The methods of complex network analysis have been developed in natural sciences (see, e.g., [22]), and later these methods have been successfully adapted to the economic and social sciences [23]. The statistical methods of complex network analysis provide new insights into the structure of economic systems, including global trade networks. The standardised network indicators are suitable for characterising the network structure regarding symmetry, clustering, density, centralisation and other aspects [24]. Countries or regions are usually represented by nodes (or vertices), and the various features of international trade by the links between the nodes. These links may be treated either as unweighted connections or weighted connections where the attached weights refer to trade volumes, values or some other relevant feature of the trade relationship.

The literature on trade network analysis is very rich. Following the applications of statistical methods for the analysis of complex networks in physics [22], similar methods have been successfully applied to problems in economics, including the analysis of the global trade network, which will be briefly reviewed below.

Many papers have applied complex network measures to analyse various patterns of global trade, including export and import flows, market integration and price transmission, supply chain management and risk management [24–37]. The most frequently used measures of social network analysis are the in- and out-degrees, the length of the shortest path (geodesic path), the density of the network, degree-centrality, betweenness centrality, closeness-centrality, eigenvalue centrality, spanning trees and the clustering of the nodes by modularities [23,38]. The set of indicators is very large. The measures applied in the present research will be discussed in the Methodology section.

The previous literature about trade network analysis deals with overall trade or trade of various products, applying various trade indicators to characterise the network. The strength of trade connection is usually measured by either the value of net export (i.e., the difference between export and import), the value of total trade (i.e., the sum of export and import), or the value of trade flow (either export or import) between countries.

The aggregate trade flows are measured by Bhattacharya et al. [36], who analyse global trade flows using the sum of the dollar value of export and import as the weight (or strength) of the connection between any two countries. Li et al. [37] use the export and import value time series for countries and compute their correlations to find out if synchronisation of the economic cycles existed between countries.

Serrano et al. [35] build their analysis of the global trade network on net export (i.e., the difference between export and import) values between countries for 1948–2000. They assess the structure of the trade network by the degrees of the nodes representing countries, i.e., the number of their connections and the total sum of their net trade values. They use a directed-graph representation and allocate the net trade values as weights to the links between countries. They measure the distribution of these weights (trade flows) using it to measure the heterogeneity of the network, establishing that 15% of all trade connections between 97–99% of all countries carry 79% of all net trade flows in 1960 and 84% in 2000.

Liu et al. [29] and Gönçer-Demiral and İnce-Yenilmez [25] also analyse international trade based on the value of net export, using centrality measures. Walther [34] analyses trade networks in West Africa, applying shortest-path computations and centrality measures to identify core actors in trade networks. The findings by Liu et al. [29] for 65 countries and 1782–2222 trade links in 2000–2016 show that closeness centrality and betweenness centrality are the highest for the same top countries. Network density and degree centrality increased, indicating increasing trade connections and higher concentration in a network reduced to the top two trade partners for the assessed countries. The modularity computations defined six to seven clusters of countries, which were found to be highly unstable. Gönçer-Demiral and İnce-Yenilmez [25] assessed the top 50 countries in 2019–2020, representing more than 90% of the total global export volume. Their modularity analysis resulted in four country clusters in 2019 and five clusters in 2020, with a more fragmented structure, probably due to the detrimental effects of the COVID-19 pandemic. Their findings show that economically strong countries tend to have the highest out-degree centralities while weak nations have the highest in-degree centralities. Betweenness centralities and closeness centralities resulted in different rankings of the 50 countries, while the network proved to be a scale-free network with a heterogeneous structure. The geographical proximity was found to be of major importance regarding international trade, and intra-cluster connections were found to be more intense than inter-cluster ones.

De Benedictis and Tajoli [24] analysed the international trade network measuring the export values of a country to its two most important trade partners. They apply network statistics such as centrality and density measures, comparing network statistics to traditional trade statistics, and point out that while there are many similarities between these two approaches, the network centrality indicators can reflect the whole network structure and not only the positions of specific countries or country pairs compared to each other. They also distinguish between local centrality measures (degree centrality and strength centrality) and global centrality measures (closeness centrality and betweenness centrality). They apply these methods for the markets of various products, such as bananas, oil, cement, footwear, movies and engines.

Ji and Fan [31] analysed the crude oil market integration using annual price changes in different markets as the basic indicator, and then the correlation between price change series of different markets was applied as a measure of integration. Using these correlations, the authors transformed the values to define metric distances attached to the links between countries. Then the minimal spanning tree was constructed to identify price transmission patterns, and centrality measures were computed to identify core and peripheral country groups.

Fair et al. [39] analysed 26 years of wheat trade data to construct a model of the wheat trade network between 1986 and 2011, focusing on the major players in the global market. They included trade linkages between countries only if the trade was sustained over at least three years, reflecting the long-term features of the network structure. Their network represents 66% of the total wheat trade volume, containing only 30% of the trade connections between 108 countries engaged in 363 partnerships. The authors analysed the time evolution of the wheat trade network and found a low level of network density and symmetry which both indicate poor resilience of the network against shocks—as was found after the financial crisis in 2008.

Sikos and Meirmanova [27] analysed the global wheat trade network of 94 countries with more than 1000 trade connections, assessing the dollar value of export and import flows between them. They used wheat trade data from 2018 and 2019 and computed node degrees and centrality measures, network density, average path length and modularity to categorise countries according to their importance in the global wheat trade network, either by the number of their trade partners or by the total trade value associated with them. The strength of the country connections was measured by net export, i.e., the difference between export and import values. Their findings showed similar or somewhat lower densities and average path lengths as in [39].

Raj et al. [40] also used social network analysis to assess the wheat supply chain in 2018 for 214 countries and 7931 trade connections. Their findings show that the six countries having the highest betweenness centralities account for more than half of the global wheat export volume, playing crucial roles in the wheat supply chain network, with implications on undernourishment.

Pacini et al. [26] analyse a specific international market, that of plastic scrap in the year 2018 for 111 countries and 1369 trade connections. The links between trade partner countries are weighted by the value of total trade (i.e., the sum of export and import). The analysis deals with centrality measures and identifies clusters of countries linked together by stronger trade ties. Betweenness centralities are computed to determine countries representing important bridges between regional markets and degree centralities and closeness centralities to identify key actors in trade clusters. The analysis showed that the countries having the highest betweenness centralities usually differ from those having the highest closeness centralities or degree centralities.

Pu et al. [28] measured the structure of global recycling trade for 189 countries and 26 industries from 1990 to 2015. They use the network density and average path length, node degrees and node strengths (weights) to analyse the network structure and compute modularities to define country clusters. Instead of using all trade connections, they work with a reduced structure handling only the top ten and top one trade partners for each country. Centrality measures are used to identify the key actors.

Nuss et al. [32] evaluate the risk of supply chains using centrality measures, while Wagner and Neshat [41] analyse supply chain vulnerability, linking countries as they are related in the supply chain and eliciting vulnerability drivers. Then they compute the correlations between these vulnerability driver indicators of various countries and apply network statistics for these to describe how the vulnerability connects different countries.

Caldarelli et al. [42] analyse the trade flows for 129 countries and 772 products. They use a different approach to computing the strength of connections between countries. Two sets of nodes are defined, one set for the countries and another for the products. For a particular country and product, the RCA (Revealed Comparative Advantage) index is computed, and then a connection is defined between a country and a product if the relevant RCA value is higher than a pre-defined threshold. Based on this network structure, centrality measures and minimal spanning trees are defined to reveal country and product clusters.

The construction of minimal spanning trees is often applied to reduce the complexity of a network. A network of N nodes can theoretically contain as much as $N \times (N - 1)/2$ connections, while the construction of MST can reduce it to the $(N - 1)$ most important non-redundant connections [43]. MST analysis has been applied for traffic networks [44]

and for various fields of economics, for example, to analyse the clustering behaviour of financial markets [43,45–47], to establish strong correspondence between the business sector and cluster structure, and to illustrate the ability of the MST methodology to convey meaningful economic information. Another interesting application was made for assessing the price transmission mechanism of the crude oil market [31].

Networks of individual commodities often show structurally different traits compared to the global network of overall trade. Therefore, individual commodity trade networks may yield unique insights [39]. The coffee green bean is a crucial export product for many tropical and subtropical countries. Its trade linkages have a considerable impact on the welfare of more than 50 developing countries and 12.5 million coffee farmers, as well as their export revenues facilitating import expense coverage and food supply [4]. Utrilla-Catalan et al. [2] analysed the green coffee trade data for the period 1995–2018 to examine the dynamics and evolution of the international market and describe the redistribution of value in the coffee supply chain, applying the tools of social network analysis. They established that during the studied period, the green coffee trade increased while the number of major actors in trade decreased, i.e., large exporting countries covered an increasing share of trade. Trade was found to be concentrated mainly on the major coffee producers, as well as on some non-producing countries, leading to greater inequality between producing and importing countries. Their analysis focused on degree-, betweenness, closeness and eigenvector centralities and modularity analysis eliciting separate clusters of countries for the studied 24 years. However, they did not look at crisis years separately, although the 2008 financial crisis was included in the time period.

The COVID-19 pandemic resulted in an enormous impact on international trade and international financial transactions, leading to an unprecedented disruption of global trade flows. The interconnection between different businesses and financial markets has greatly contributed to spreading the effects of COVID-19 on the economy and household well-being. In order to measure the impact of COVID-19 on international trade and stock markets, the intra-firm relationship between suppliers and customers and the variables of financial companies have been analysed by Zhang [48]. The effects of the pandemic on global stock market indices were analysed using complex network methods by Aslam et al. [49], with data from 15 October 2019 to 7 August 2020 identifying a significant impact of COVID-19 through structural changes in nodes, reduced connectivity and significant differences in the topological characteristics of the network. The commercial and financial dynamics were analysed by Louati et al. [50], and they established that during the COVID-19 lockdowns, the flow of information and trade transactions considerably changed, leading to changes in the supremacy of regions and crisis transmission patterns. Complex network analysis algorithms have been successfully applied to demonstrate the changes in network structures, not only in terms of trade volumes but in the number of connections and relationships [51]. The global trade structure between the pre-pandemic year 2018 and the pandemic year 2020 was compared by Coquidé et al. [51], determining significant changes in the ranking of countries from 2018 to 2020 in world trade, but the impacts differed by major product groups.

Global agricultural trade has been said to be resilient to the impacts of the COVID-19 pandemic [52]. Using a reduced-form, gravity-based econometric model for monthly trade, the actual effects of the pandemic—including those of incidence rates, government policy restrictions, limited human mobility and lockdown effect on global agricultural trade was estimated up to the end of 2020. The findings show that agricultural trade remained quite stable through the pandemic, reduced only by 5–10%, which is two to three times smaller than the estimated impact on trade in the non-agricultural sector. However, non-food agricultural trade was most severely impacted by the pandemic, while the effects on the majority of food and bulk agricultural commodities were insignificant [52]. The sustainability and resilience of the agricultural and food supply chains is a topical issue. It refers to the capability of the supply network to manage and mitigate disruptions due to various reasons such as natural disasters, climate change or human-caused shocks,

i.e., any uncontrollable event. Agricultural supply chains are exposed to conditions where disruptions occur. These networks are highly vulnerable to events caused by abrupt changes in climate and aggravated regional geography. Resilience can be assessed by way of resilience metrics related to availability and connectivity, based on the simulation of disruptive events and identify resilient designs using mathematical programming, as in the analysis of the Colombian coffee supply chain [53] during the COVID-19 pandemic. Resilience represents the speed at which a system returns to equilibrium after a disturbance. Resilience can be modelled by way of graph theory or by mathematical programming using mixed linear models [53,54]. The COVID-19 pandemic did not change coffee consumption significantly in Europe, America or Asia, as the close-down has channelled consumption towards homes and other confined environments [55].

The motivation of the present research is to analyse the international network of coffee green bean trade, focusing on the year 2020, the year of the COVID-19 pandemic, following the example of [8,51]. Our aim is to find out if the global crisis had significant impacts on the global coffee trade network. More specifically, the specific questions we are addressing are:

- Q1: Has the list of largest actors changed between the pre-COVID 2018 and post-COVID 2020 years?
- Q2: Has the pandemic differently impacted exporters and importers?
- Q3: Have trading group structures changed?
- Q4: Have core and periphery countries, or big and small players, been affected differently?

Naturally, trade volumes decreased due to restrictions on transportation and also difficulties of production related to restricted labour movements. We did not directly compare the absolute trade volumes but focused on the structural features of the trade networks of the two analysed years. Therefore, trade flows between countries were assessed as percentages of the global trade value of the particular year, and these shares were used for comparisons across years. This treatment makes the price index adjustments unnecessary in our computations. The majority of papers dealing with the impacts of COVID-19 on trade networks of specific commodities either dealt with the leading countries of the trade network or with a specific group of countries (see [51] for examples). The present research, however, deals with all trade relationships in the international coffee market in 2018 and in 2020.

For this purpose, similar to the approach used in [8] and [51], the results of 2020 are contrasted to the results of 2018, the last "normal" year before the outbreak of the pandemic. The main question of the analysis was the changes in the trade network structure due to the pandemic, the relative roles and importance of exporting and importing regions, the resilience in the connectedness of closely-knit trading communities, and the identification of the most vulnerable actors in the network. Findings show that from 2018 to 2020, the role of some major actors considerably changed, many trade relationships were disrupted, and the number of trade connections also changed, with some winners and many losers. The study also demonstrated how the network measures are suitable to quantify these changes and identify differences between affected countries.

The structure of the paper is as follows: Section 2 describes the data used for the analysis, presents the main research objectives and motivations, and gives the main network metrics that are used for the evaluation. Section 3 presents the results of the analysis, including network structure, centrality measures, trading communities and minimum spanning trees for 2018 and 2020. Section 4 contains a discussion of our results, and Section 5 draws conclusions, presents the main implications and points out the limitations of the research. Finally, Appendix A gives the list of countries with their three-letter country codes used in the paper.

## 2. Materials and Methods

### 2.1. Data

The dataset for the present analysis was downloaded from the publicly available database of Trade Map ITC-Trade statistics [56] for international business development for

the year 2001–2020 under the Harmonized System HS6, code 090111, 'Coffee; not roasted or decaffeinated"—using the 1992 revision of the Harmonized System for 6-digits [57] The time period was chosen because of the availability of data. The present paper focuses only on two years, 2018 and 2020, from the dataset. The downloaded dataset includes, among others, the main coffee producers: Brazil, Colombia, Vietnam, Guatemala, Honduras, Mexico, Indonesia, Peru, India, Costa Rica, Ethiopia, El Salvador, Côte d'Ivoire, Nicaragua, Kenya and Uganda, while the consumer side included the United States of America, Belgium and Germany, among others. The trade data have been selected for trade relationships with exporter countries whose share in global green bean coffee exports is not less than 1% of the global value; these trade flows contributed approximately 90% of the total export value of green bean coffee in 2020. The data are the US dollar value of green bean coffee exports from the world's major exporting countries. The dataset contains 234 countries /trading areas and 1190 trade relations in 2018 and 1160 trade relations in 2020.

As our analysis refers to two years, price levels did change from the first one to the second. Therefore, it would be reasonable to deflate 2020 trade values with the relevant price index. Trade values are reported in dollar values that take into account the exchange rates of the various countries. These incorporate price level differences between countries within the same year. Therefore, to correct for inflation between 2018 and 2020, it would be sufficient to use the same price index for all countries as an aggregated international price index. However, as it will be shown later in the analysis, trade relationships are analysed within the same year, and trade values are not directly compared across years. Thus within-year comparisons are not affected by the actual price index between the two years. Across-year comparisons are only made based on the weighted network metrics, but these are invariant to the actual values of the network flows and depend only on their proportions. This means that when multiplying each trade value within the same network by a constant (i.e., the 2020 trade values by the price index of 2020 to 2018), the network parameters remain the same and will not influence our results and conclusions. Therefore, avoiding the deflation of the trade values does not affect our results and conclusions.

In the coffee trade dataset, the same trade value may be reported differently by the exporter and the importer, taking into account the differences in CIF and FOB costs and the reliability of the reporter countries. Throughout the paper, we often refer to trade values as 'export values', as is done in [8], but the data were checked to ensure that flows are similarly reported by the exporter and the importer. As we are using the trade percentage distribution and not the absolute dollar values, the difference between reported export and import values will not distort our results considerably.

In order to provide more general information regarding the green bean coffee trade, its temporal evolution (Figure 1) and country shares (Figure 2) are provided.

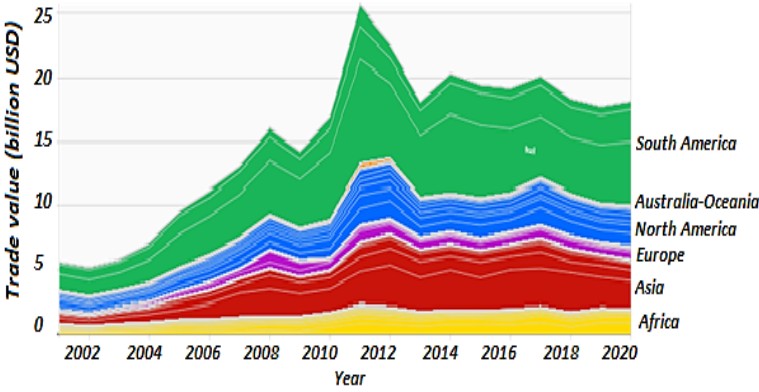

**Figure 1.** Stacked Green Bean Coffee Exports for the period 2001–2020 by continent. Source: Authors' own construction based on publicly available data from [56].

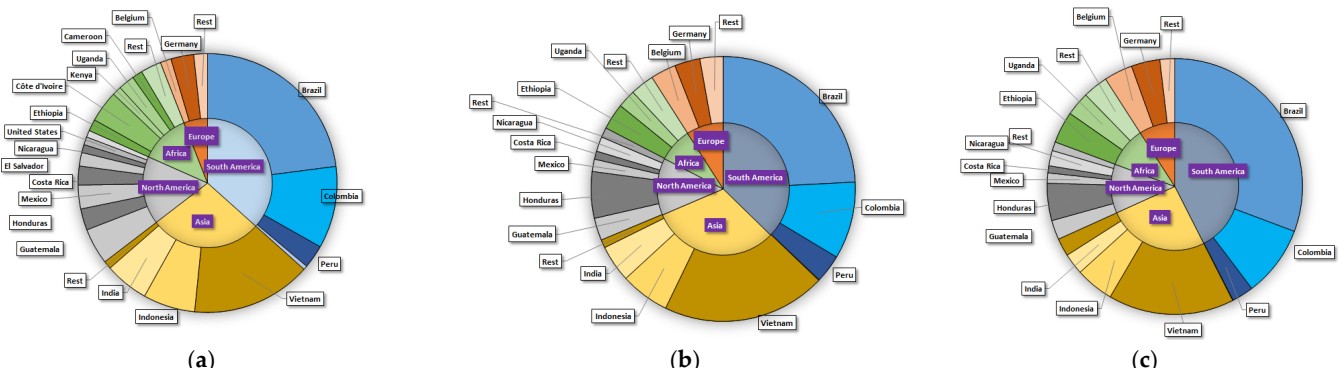

**Figure 2.** Distribution of coffee exports: (**a**) Year 2001; (**b**) Year 2018; (**c**) Year 2020. Source: Authors' own construction based on publicly available data from [56].

Figure 1 depicts that the value of the green bean coffee trade nearly steadily increased from year to year. The largest contribution came from countries in South America (green in Figure 1), namely Brazil and Colombia, both of which have a contribution of about 30%. Furthermore, countries in Asia (red) and North America (blue) contribute approximately 18% and 14%, respectively. After reaching its peak value in 2008, the value of the green bean coffee trade declined because of the global financial crisis, while in 2012, the decrease was caused by the European Union crisis [2]. Figure 2 demonstrates which countries have an important role in the green bean coffee trade in each region.

In 2001, among African producers, Cameroon, Côte d'Ivoire and Tanzania had significant roles in green bean coffee export, but that role was vanishing year by year until 2020. Kenya has also experienced a decline in terms of exports though a little less than the formerly mentioned countries. What remains is Ethiopia and Uganda, which still play an important role in the export of coffee from Africa. A similar pattern is seen in North America. El Salvador and the USA no longer have a dominant role in green bean coffee export compared to 2001. The significant exporters in 2020 in North America are Honduras, Guatemala, Nicaragua, Mexico and Costa Rica.

In South America and Europe, there has been no significant change regarding the dominant role in green bean coffee export. Brazil, Colombia and Peru are still leading in South America, while Germany and Belgium are still leading in Europe. China appears as a new competitor in green bean coffee export from Asia and Papua New Guinea from Australia/Oceania.

### 2.2. Metrics of Social Network Analysis

The trade network was analysed using the statistical indicators of social network analysis [2]. The following indicators are applied based on [23,24,26,29,38].

A graph is constructed in a way that countries are represented by the nodes of the graph. The linkages between countries are represented by the edges between nodes: $e(i,j)$ denotes the edge between countries i and j ($i,j = 1 \ldots N$, where N denotes the number of countries). The edges can have weights associated with them (representing the distance between countries or the strength of the connection between countries): $w(i,j)$ is the weight associated with $e(i,j)$. If the relationship between countries is symmetric (undirected edges), then the maximum number of edges is $N \times (N - 1)/2$, while for a directed graph (where the direction of the connection between countries is important), the maximum number of edges is $N \times (N - 1)$. A numeric representation of edges is the adjacency matrix $A = [a(i,j)]$, where $a(i,j) = 1$ if there is a link from node i to node j, and $a(i,j) = 0$ otherwise ($i,j = 1 \ldots N$). For undirected graphs $a(i,j) = a(j,i)$, while for directed graphs, these may differ.

The degree of a node (country) is the number of edges entering the country:

- for undirected graphs: $k(i) = \Sigma_{(j=1 \ldots N, j \neq i)} a(i,j)$.
- For directed graphs

- k(i, in) is the number of incoming edges to country i, and
- k(i, out) is the number of outgoing edges from country i.

The strength of a node (country) is the sum of weights of edges entering the country:

- for undirected graphs: $s(i) = \Sigma_{(j=1 \ldots N, j \neq i)} w(i,j)$.
- For directed graphs
  - s(i, in) is the sum of weights of the incoming edges to country i, and
  - s(i, out) is the sum of weights of outgoing edges from country i.

The density of a graph is the number of its edges compared to the maximum number of edges possible, i.e., for undirected graphs, it is $D = [\Sigma_{(i,j=1 \ldots N)} a(i,j)]/[N \times (N-1)/2]$, while for directed graphs is $D = [\Sigma_{(i,j=1 \ldots N)} a(i,j)]/[N \times (N-1)]$. Similarly, the average weighted density (also called average strength) is $S = [\Sigma_{(i,j=1 \ldots N)} w(i,j)]/[N \times (N-1)/2]$ for undirected graphs and $S = [\Sigma_{(i,j=1 \ldots N)} w(i,j)]/[N \times (N-1)]$ for directed ones.

The length of a path from node i to node j is the number of edges connecting them, while the weighted length is the sum of weights attached to the edges connecting the two countries. The shortest path between two nodes is called *geodesics or geodesic path*, and it is an important problem in graph theory to determine the shortest path between two nodes. The shortest weighted path is the path with the minimum weighted length between two nodes. By denoting d(i,j) the shortest path length from node i to node j (and d(i,i) = 0 by definition), the average path length of a network is $APL = [\Sigma_{(i,j=1 \ldots N)} d(i,j)]/[N \times (N-1)]$ (assuming that the network is connected, i.e., any node can be reached from any other nodes). APL is the smallest when each node is directly connected to any other node, and then APL = 1. For disconnected graphs, the unreachable nodes are left out of the computation, and the formula is adjusted accordingly.

Degree centrality is the simplest centrality concept that ranks country i in a network according to the number of connections it has with other countries. A node (country) with a degree k(i) = N − 1 would be directly connected to all other countries in the network, hence quite central to the network. The degree centrality of a country node is simply CD(i) = k(i)/(N − 1) so that it ranges from 0 to 1 and indicates what share a node has compared to the total number of all possible direct connections. Similarly, in-degree centrality CD(i, in) and out-degree centrality can be defined as CD(i, in) = k(i, in)/(N−1) and CD(i, out) = k(i, out)/(N − 1). The percentage values of strength centralities (in- and out) can be computed if the division is not by (N−1) but by the total trade flows in the network [3], resulting in the average share of the total trade flows. Weighted degree centralities use the sum of edge weights linked to the node instead of node degrees.

Betweenness centrality indicates how important a country is in terms of connecting other countries. It can be understood as a measure of centrality in a graph based on the shortest paths:

- P(i,k,j) represents the number of all the shortest paths between nodes k and j that contain node I.
- P(k,j) denotes the total number of shortest paths between countries k and j. It is then possible to estimate how important country i is in connecting k and j by the ratio P(i,k,j)/P(k,j). Note, that $P(k,j) = \Sigma_{(i=1 \ldots N, i \neq k,j)} P(i,k,j)$. If the ratio is near 1, then i lies on most of the shortest paths connecting countries k to j, while if it is close to 0, then i is less important in connecting k and j.
- When averaging across all pairs of nodes, the betweenness centrality of country i is defined as $CB(i) = \Sigma_{\{k,j=1 \ldots N, k \neq i, k \neq j, j \neq i\}} [P(i,k,j)/P(k,j)]/[(N-1) \times (N-2)/2]$. Betweenness centrality shows the importance of a country in connecting to other countries. A country with high betweenness centrality holds a powerful role in the network as a bottleneck.

Closeness centrality [58,59] measures how easily a country can reach other countries in the network, i.e., how close a node is to any other node in the network. It is measured by the inverse of the average distance between nodes i and j:

- $CC(i) = (N - 1)/\Sigma_{\{j=1 \ldots N, j \neq i\}} \, l(i,j)$, where $l(i,j)$ is the number of links (edges) in the shortest path between i and j. The higher the closeness centrality score is, the more central—or closer—a node is to all other nodes in the network.
- The weighted version of closeness centrality is [3]: if $w(i,j)$ is the weight of the edge $e(i,j)$, then: the average weight (e.g., the average bilateral trade volume in world trade) is:

$$U(i,j) = N \times w(i,j)/[\, \Sigma_{\{k=1 \ldots N\}} \, (\Sigma_{\{l=1 \ldots N\}} \, w(k,l))],$$

Then the weighted geodesic distance over the i to j path of $(i–z_1–z_2–z_3 \ldots –z_{n-2}–j)$ is:

$$d(i,j) = \min \, [1/U(i,z_1) + 1/U(z_1,z_2) + \ldots 1/U(z_{n-3},z_{n-2}) + \ldots 1/U(z_{n-2},j)]$$

Then the weighted closeness centrality (for directed networks) for node i is: $CCW(i, out) = (N - 1)/[\Sigma_{\{j=1 \ldots N, j \neq i\}} \, d(i,j)]$ and $CCW(i, in) = (N-1)/ \, [\Sigma_{\{j=1 \ldots N, j \neq i\}} \, d(j,i)]$. The index can be interpreted as the inverse of the average weighted geodesic distance from i to its $(N - 1)$ potential connections. As an example, a value of $CCW(i, in) = 0.5$ means that node i is $1/0.5 = 2$ units away from the other nodes, where one unit is the average bilateral trade flow in world trade.

Betweenness centrality measures how much a country acts as an intermediary or gatekeeper in the trade network. Both Degree centrality and Closeness centrality (CC) are based on the idea that the centrality of a node in a network is related to its distance to the other nodes, while Betweenness Centrality (BC) is based on the idea that central nodes stand between others.

Eigenvector centrality [40] measures how well each node (country) is connected to other influential nodes (countries), the computing power and status of respective countries and their connections, giving higher eigenvector centrality values to nodes whose connecting nodes also have high centralities. Eigenvector centrality can be useful in identifying important secondary markets, importing raw products and selling processed products. These secondary markets have to be well-connected to influential countries to maintain their status. The eigenvector centrality of node i is equivalent to the sum of the centralities of its neighbours:

$$EC(i) = (1/\lambda) \times, \Sigma_{\{j=1 \ldots N\}} \, [a(i,j) \times EC(j)],$$

where

- $\lambda$ is the largest eigenvalue of the adjacency matrix $A = [a(i,j)]$ and
- $E = [EC(1), EC(2), \ldots EC(N)]$ is the eigenvector belonging to the eigenvalue $\lambda$.

Social network analysis tools are also suitable for identifying node clusters or communities. A widely used measure for community decomposition is modularity, which measures the density of links inside communities as compared to links between communities. The value of the modularity lies in the range $[-1, 1]$, where the closer the value to 1, the better the quality of the partitions. Modularity is defined as:

$$Q = [1/2m] \times \sum_{\{i,j=1 \ldots N\}} [w(i,j) - (s(i) \times s(j) /2m] \times \delta(c(i), c(j)),$$

where

- $c(i)$ represents the community that contains node I, and
- $\delta(c(i), c(j)) = 1$ if $c(i) = c(j))$, and $\delta(c(i), c(j)) = 0$ otherwise,
- $w(i,j)$ is the edge weight between nodes i and j,
- $s(i) = \sum_{\{j=1 \ldots N\}} w(i,j)$, the sum of weights of the edges connected to node i,
- $m = (1/2) \times \sum_{\{i=1 \ldots N\}} s(i) = \sum_{\{i,j=1 \ldots N\}} w(i,j)$, the total edge weight of the network.

The concept of Minimum Spanning Tree (MST) is defined next. A network is connected if from any node there exists a path to any other node. A tree is a connected network that has no cycles or loops (i.e., no paths where the starting node is the same as the ending node). A network of N nodes is a tree if and only if it has exactly $(N - 1)$ edges. A forest is a network in which each component is a tree. The Minimum Spanning Tree (MST) is a tree

(a connected graph without loops, containing N − 1 edges) such that the sum of all weights (distances) is minimised. The advantage of constructing MST in a network is that it can greatly reduce the complexity of the network because instead of the theoretically possible N × (N − 1)/2 connections, it can reduce the links to only the N − 1 most important non-redundant connections in a graphical manner. The MST approach also provides useful information in terms of the centrality or otherwise of individual equity markets (nodes) in the overall system [43]. Let's denote by R(i,j) the shortest distance in the MST between nodes i and j, then the farness of node i can be defined as Farness(i)= $\Sigma_{\{j=1 \dots N, , j \neq i\}}$ R(i,j), referring to the average distance (weight) of i compared to all other nodes. This concept can be used to define clusters of nodes.

The network analysis in the present paper was performed by the GEPHI software, a freely available software particularly useful for social network analysis, favoured or its flexible capacities for network visualisation and supply of many network measures [60,61]. This software was applied in many trade network analysis papers [25–29,35]. Gephi can compute the above centrality measures and uses a modularity optimisation method to decompose a gigantic network into several relatively independent modules (groups, clusters), i.e., sets of highly connected nodes.

### 2.3. Theoretical Background and Methodology for Addressing the Research Questions

Our paper analyses the international coffee bean trade structure using the above network analysis measures and compares the pre-COVID year 2018 to that of the year 2020, in which the impacts of the pandemic were seriously felt. We compare the indicators of these two years to identify crucial changes due to the pandemic in disrupting former trade linkages and establishing new trade positions. The four research questions of our research are analysed by the following network methods:

Q1: Have the largest actors changed between the pre-COVID and post-COVID years?

The largest actors of a network can be identified either by the number of their trading partners or by their share in the total trade value of the network. To examine the first option, we compute the in-degrees of the countries, with large in-degrees indicating countries with many import relationships. In order to examine the second option, the weighted degrees are computed. Large weighted in-degrees (expressed as % of the total trade value) indicate high shares of importer countries in total trade, while large weighted out-degrees indicate high shares of exporter countries in total trade. Countries with large in-degrees and large weighted in-degrees are large importers, while countries with large out-degrees and weighted out-degrees are large exporters. The list of such countries is compared for 2018 and 2020 to see any shifts or significant changes.

Q2: Has the pandemic differently impacted exporters and importers?

As was established above, the countries ranking high by in-degrees and weighted in-degrees are major importers; countries ranking high by their out-degrees and weighted out-degrees are major exporters. Their shares, i.e., %-values of weighted degrees, are compared between the two years to see if there is any change and whether the change differs for importers and for exporters.

Q3: Are trading group structures changed?

Trading groups can be determined by modularity computation in network analysis, which identifies close trading communities based on the trade flows between countries. The cluster structures between the two years are compared, and the network characteristics, such as density, average path length and network diameter, are computed and compared between the two years.

Q4: Have core and periphery countries been affected differently?

Core and periphery countries can be identified using the minimum spanning tree approach or the centrality index values. The minimum spanning tree is a kind of backbone for the network, where only the most important linkages are kept—i.e., those links that ensure that each country participates in the trade flow, possibly being linked to its most important trade partners. The core countries are those having many partners and large

trade values in the minimum spanning tree, and peripheral countries are those with low-value links and few partners. The other aspect of core and periphery relations is the centrality distribution. Countries with betweenness centralities have a certain controlling role in the network, i.e., if they may disappear from the trade network (e.g. due to a natural disaster or a human-induced catastrophe like a war), the network will break up. Countries with large closeness centrality, on the other hand, have strong relations with direct neighbours; therefore, they are less dependent on the control of countries of high betweenness centrality. Finally, countries with high eigenvector centrality are connected to the powerful actors of the network, though they may not be powerful traders in themselves. The changes in the centrality distributions can also indicate shifts in the positions of core and periphery countries.

## 3. Results

### 3.1. Comparison of Trade Network Structures of 2018 and 2020

Figure 3 shows that the most outstanding trade actors are Brazil, the USA, Germany, Colombia, Vietnam and Belgium, i.e., major producers/exporters and major importers play crucial roles in the trade network. Simple visual analysis shows, however, that there are changes between the two studied years: while Brazil, Vietnam, the USA and Germany remain strong players in the network, the role of Ethiopia, China and Mexico considerably decreased by 2020. Degree distributions underline this visual statement, especially the weighted degree values, although weighted degrees also show a considerable weakening of Vietnam (Tables 1 and 2).

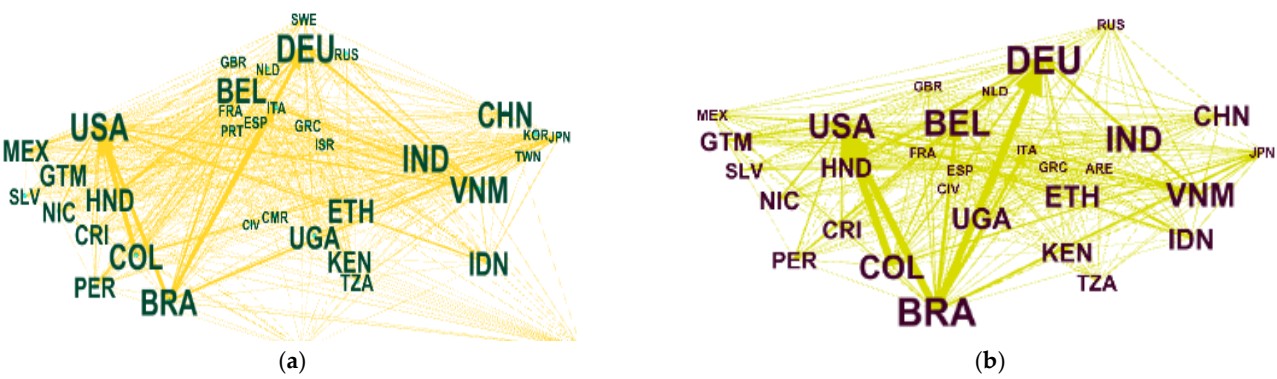

(**a**)          (**b**)

**Figure 3.** Coffee Trade network in 2020—label size is proportional to the number of trade partners, and edge thickness is proportional to trade value: (**a**) Year 2018; (**b**) Year 2020. Note: Only the countries with at least 20 trade partners are shown. Country codes and names are presented in Appendix A.

**Table 1.** Degree distributions—top 30 values, 2018 and 2020.

| In-Degree 2018 | | Out-Degree 2018 | | Degree, 2018 | | In-Degree 2020 | | Out-Degree 2020 | | Degree 2020 | |
|---|---|---|---|---|---|---|---|---|---|---|---|
| ESP | 22 | BRA | 88 | BRA | 90 | ESP | 21 | BRA | 93 | BRA | 96 |
| FRA | 22 | COL | 70 | DEU | 90 | ARE | 20 | COL | 75 | DEU | 95 |
| GRC | 22 | ETH | 70 | USA | 90 | DEU | 20 | DEU | 75 | BEL | 86 |
| ITA | 22 | IND | 70 | IND | 84 | FRA | 20 | IND | 72 | IND | 85 |
| BEL | 21 | USA | 70 | BEL | 77 | GBR | 20 | ETH | 68 | USA | 80 |
| DEU | 21 | DEU | 69 | VNM | 75 | GRC | 20 | BEL | 67 | COL | 78 |
| GBR | 21 | VNM | 64 | COL | 74 | ITA | 20 | VNM | 65 | VNM | 78 |
| ISR | 21 | UGA | 63 | CHN | 70 | JPN | 20 | UGA | 63 | ETH | 68 |
| NLD | 21 | IDN | 58 | ETH | 70 | NLD | 20 | USA | 61 | IDN | 65 |
| RUS | 21 | HND | 57 | IDN | 67 | RUS | 20 | IDN | 58 | UGA | 65 |
| AUS | 20 | BEL | 56 | UGA | 64 | AUS | 19 | HND | 55 | CHN | 63 |
| JPN | 20 | KEN | 56 | HND | 60 | BEL | 19 | NIC | 54 | KEN | 57 |

**Table 1.** *Cont.*

| In-Degree 2018 | | Out-Degree 2018 | | Degree, 2018 | | In-Degree 2020 | | Out-Degree 2020 | | Degree 2020 | |
|---|---|---|---|---|---|---|---|---|---|---|---|
| KOR | 20 | GTM | 54 | KEN | 60 | CAN | 19 | GTM | 52 | GTM | 55 |
| PRT | 20 | CHN | 51 | GTM | 55 | CHN | 19 | KEN | 50 | HND | 55 |
| SWE | 20 | NIC | 50 | MEX | 55 | DNK | 19 | CRI | 47 | NIC | 55 |
| TWN | 20 | PER | 48 | NIC | 50 | KOR | 19 | PER | 46 | CRI | 52 |
| USA | 20 | CRI | 42 | PER | 50 | SGP | 19 | TZA | 45 | PER | 49 |
| ZAF | 20 | TZA | 40 | CRI | 48 | TWN | 19 | CHN | 44 | TZA | 45 |
| ARE | 19 | MEX | 39 | TZA | 40 | USA | 19 | SLV | 38 | SLV | 41 |
| CAN | 19 | SLV | 35 | SLV | 36 | FIN | 18 | CIV | 21 | MEX | 25 |
| CHE | 19 | CMR | 21 | ESP | 22 | ISR | 18 | MEX | 11 | CIV | 22 |
| CHN | 19 | CIV | 19 | FRA | 22 | SAU | 18 | | | ESP | 21 |
| SGP | 19 | | | GRC | 22 | SWE | 18 | | | ARE | 20 |
| DNK | 18 | | | ITA | 22 | ZAF | 18 | | | FRA | 20 |
| FIN | 18 | | | CMR | 21 | HKG | 17 | | | GBR | 20 |
| LVA | 18 | | | GBR | 21 | MYS | 17 | | | GRC | 20 |
| NZL | 18 | | | ISR | 21 | NOR | 17 | | | ITA | 20 |
| POL | 18 | | | NLD | 21 | NZL | 17 | | | JPN | 20 |
| SAU | 18 | | | RUS | 21 | PRT | 17 | | | NLD | 20 |
| TUR | 18 | | | AUS | 20 | TUR | 17 | | | RUS | 20 |

**Table 2.** Weighted degree distributions as % of total trade value—top 30 values.

| Weighted In-Degree, 2018 | | Weighted Out-Degree, 2018 | | Weighted Degree, 2018 | | Weighted In-Degree, 2020 | | Weighted Out-Degree, 2020 | | Weighted Degree, 2020 | |
|---|---|---|---|---|---|---|---|---|---|---|---|
| USA | 21.5% | BRA | 24.7% | BRA | 24.7% | USA | 21.8% | BRA | 29.2% | BRA | 29.2% |
| DEU | 13.4% | VNM | 16.0% | USA | 22.4% | DEU | 14.5% | COL | 14.4% | USA | 22.7% |
| ITA | 7.5% | COL | 12.9% | DEU | 16.6% | BEL | 7.8% | VNM | 11.1% | DEU | 17.7% |
| BEL | 7.4% | HND | 6.3% | VNM | 16.2% | ITA | 7.1% | HND | 5.1% | COL | 14.9% |
| JPN | 6.0% | IDN | 4.6% | COL | 13.4% | JPN | 5.7% | IDN | 4.7% | BEL | 11.4% |
| CAN | 3.4% | ETH | 4.5% | BEL | 10.6% | CAN | 3.5% | ETH | 4.7% | VNM | 11.3% |
| NLD | 3.3% | GTM | 3.9% | ITA | 7.5% | FRA | 3.2% | GTM | 3.8% | ITA | 7.1% |
| FRA | 3.0% | PER | 3.8% | HND | 6.9% | NLD | 3.1% | PER | 3.8% | JPN | 5.7% |
| ESP | 2.8% | BEL | 3.3% | JPN | 6.0% | ESP | 2.7% | BEL | 3.6% | HND | 5.1% |
| GBR | 2.6% | DEU | 3.1% | IDN | 5.3% | KOR | 2.6% | DEU | 3.2% | IDN | 4.9% |
| KOR | 2.2% | IND | 2.9% | ETH | 4.5% | GBR | 2.1% | UGA | 3.0% | ETH | 4.7% |
| RUS | 1.6% | UGA | 2.5% | GTM | 3.9% | RUS | 1.9% | IND | 2.7% | GTM | 3.8% |
| POL | 1.6% | NIC | 2.4% | PER | 3.8% | POL | 1.5% | NIC | 2.6% | PER | 3.8% |
| SWE | 1.5% | CRI | 1.8% | IND | 3.6% | SWE | 1.5% | CRI | 1.9% | CAN | 3.5% |
| AUS | 1.3% | MEX | 1.8% | CAN | 3.4% | AUS | 1.3% | MEX | 1.4% | IND | 3.3% |
| DZA | 1.2% | KEN | 1.3% | NLD | 3.3% | MYS | 1.1% | KEN | 1.2% | FRA | 3.2% |
| MYS | 1.0% | CHN | 1.1% | FRA | 3.0% | SAU | 1.1% | USA | 0.9% | NLD | 3.1% |
| FIN | 1.0% | USA | 0.9% | ESP | 2.8% | DZA | 1.1% | TZA | 0.8% | UGA | 3.0% |
| CHN | 0.9% | TZA | 0.8% | GBR | 2.6% | FIN | 1.0% | CHN | 0.7% | ESP | 2.7% |
| SAU | 0.8% | CIV | 0.8% | UGA | 2.5% | TUR | 1.0% | SLV | 0.6% | NIC | 2.6% |
| TUR | 0.8% | SLV | 0.6% | MEX | 2.4% | CHN | 0.9% | CIV | 0.5% | KOR | 2.6% |
| GRC | 0.7% | CMR | 0.2% | NIC | 2.4% | TWN | 0.7% | | | MEX | 2.1% |
| IND | 0.7% | | | KOR | 2.2% | GRC | 0.7% | | | GBR | 2.1% |
| IDN | 0.7% | | | CHN | 2.0% | MEX | 0.7% | | | CRI | 2.0% |
| MEX | 0.6% | | | CRI | 1.8% | SDN | 0.7% | | | RUS | 1.9% |
| HND | 0.6% | | | RUS | 1.6% | NOR | 0.7% | | | CHN | 1.6% |
| THA | 0.6% | | | POL | 1.6% | EGY | 0.6% | | | POL | 1.5% |
| COL | 0.6% | | | SWE | 1.5% | IND | 0.6% | | | SWE | 1.5% |
| EGY | 0.5% | | | AUS | 1.3% | COL | 0.6% | | | AUS | 1.3% |
| NOR | 0.5% | | | KEN | 1.3% | JOR | 0.5% | | | KEN | 1.2% |

In 2018 the in-degrees of the top 30 countries ranged from 22 to 18, and these countries are the main importing countries. There are 22 countries with non-zero out-degrees,

i.e., having export relationships. Among them, the USA, Germany, Belgium and China are the only countries that are not major producers. Regarding the total degrees, the list of the countries with the most trade connections begins with Brazil, Germany, USA, India, Belgium, Vietnam, Colombia, China, Ethiopia and Indonesia (Table 1). Former research [2] for earlier years shows a somewhat different pattern: regarding the in-degree of different countries in the international green coffee trade, the leading countries were Belgium, the United States, the United Kingdom, Japan and Denmark, while the leaders with regard to out-degree were Vietnam, Indonesia, India and Brazil in 2002. However, our analysis shows that by 2020 little change is seen among importers: the top 20 countries are nearly the same as in 2018, with Switzerland, Latvia and Poland falling out by 2020 and Hongkong, Norway and Malaysia entering the top 20, with 21–17 trade (importer) partners.

The out-degrees show a higher concentration of exporter countries. There are only 21 countries in 2020 with positive out-degrees (i.e., export relations), and the leading ones are the major exporters—Brazil, Colombia, India, Ethiopia, Vietnam and Indonesia—but a few large importers also rank highly as exporters, too (Germany, Belgium, the USA and China). The top 10 countries include three big consumers (Germany, Belgium and the USA). The list contains the same countries as in 2018, with Cameroon being the only one present in 2018 but not in 2020.

It is worth noting that from 2018 to 2020, while the in-degrees decreased a little (importers losing some trade partners), the out-degrees increased for the top 10 exporter countries, indicating a more diversified export partnership for them. The USA, Indonesia and Ethiopia are the three important exporters actually losing trade partners by 2020. The merging of in- and out-degrees represent the total trade connections of countries, and the exporters dominate this list, although the major importers—USA, Belgium (11th in 2018) and China (14th in 2018)—are ranking higher than either in the in-list and the out-list, meaning that their activities are considerably strong both in export and import.

The weighted degree distributions are somewhat different, especially the weighted in-degree list. The weighted out-degrees are dominated by Brazil, Vietnam and Colombia in both 2018 and 2020, while the weighted in-degree list is led by the USA, Germany, Belgium, Italy, Japan, Canada, the Netherlands and France—and the list of 2018 is very similar to 2020. (Table 2). However, it is worth noting that while Brazil and Germany increased the number of their trade partners as well as their share in global trade value, the USA lost partners and lost share in global trade value as well. Vietnam, Honduras and Indonesia are losers in terms of trade value, but Vietnam actually increased the number of trade partners, while its trade value decreased from 16.2% in 2018 to 11.3% in 2020. The correlation between in-degrees and weighted in-degrees is of medium level ($r = 0.456$ in 2018 and $r = 0.448$ in 2020), while the out-degrees are highly correlated to weighted out-degrees in both years ($r = 0.721$ in 2018 and $r = 0.718$ in 2020), as well as the degrees and weighted degrees ($r = 0.784$ in 2018 and $r = 0.777$ in 2020).

*3.2. Centralities*

Centralities measure the importance of countries within the trade network. Closeness centrality measures the importance of a country in its proximity, i.e., its immediate trade partners. It shows similar patterns in 2018 and in 2020, with basically the same countries having positive closeness centrality measures (Table 3). However, the range of values widened from the 0.463–0.673 range in 2018 to the 0.434–0.724 range in 2020 (values were normalised to the 0–1 range). The ranking of the countries considerably changed, with the USA falling back from a value of 0.63 (2nd in 2018) to 0.597 (8th in 2020). The largest loss was experienced by Mexico (18th with 0.524 in 2018 and 21st with 0.434 in 2020). Brazil maintained its leading position, and Colombia considerably improved its importance.

**Table 3.** Closeness and betweenness centralities, higher than zero values (2018, 2020, normalised).

|  | Closeness Cty, 2018 | | Closeness Cty, 2020 | | Betweenness Cty, 2018 | | Betweenness Cty, 2020 | |
|---|---|---|---|---|---|---|---|---|
| 1 | BRA | 0.672897 | BRA | 0.724490 | USA | 0.010617 | USA | 0.008533 |
| 2 | USA | 0.631579 | DEU | 0.625551 | DEU | 0.008795 | DEU | 0.007230 |
| 3 | ETH | 0.629787 | COL | 0.620087 | BEL | 0.005205 | BEL | 0.005316 |
| 4 | DEU | 0.623377 | ETH | 0.600823 | IND | 0.004454 | BRA | 0.003792 |
| 5 | UGA | 0.612766 | IND | 0.599156 | KEN | 0.002999 | IND | 0.003738 |
| 6 | COL | 0.607595 | UGA | 0.599156 | BRA | 0.00256 | VNM | 0.003331 |
| 7 | IND | 0.607595 | VNM | 0.599156 | VNM | 0.002552 | KEN | 0.002657 |
| 8 | VNM | 0.592593 | USA | 0.596639 | CHN | 0.002457 | CHN | 0.002299 |
| 9 | KEN | 0.585366 | BEL | 0.589212 | IDN | 0.001696 | UGA | 0.001386 |
| 10 | IDN | 0.578313 | HND | 0.578947 | UGA | 0.001247 | COL | 0.001054 |
| 11 | CHN | 0.5625 | KEN | 0.568000 | HND | 0.001091 | CRI | 0.000884 |
| 12 | BEL | 0.55814 | NIC | 0.561265 | MEX | 0.000923 | IDN | 0.000565 |
| 13 | HND | 0.555985 | IDN | 0.559055 | COL | 0.000507 | NIC | 0.000525 |
| 14 | GTM | 0.54717 | TZA | 0.555556 | CIV | 0.000503 | CIV | 0.000492 |
| 15 | PER | 0.543396 | PER | 0.544061 | CRI | 0.000124 | GTM | 0.000459 |
| 16 | TZA | 0.539033 | GTM | 0.537879 | PER | 0.000096 | PER | 0.000397 |
| 17 | NIC | 0.536496 | CRI | 0.531835 | GTM | 0.000094 | SLV | 0.000218 |
| 18 | MEX | 0.523636 | CHN | 0.520147 | SLV | 0.000035 | MEX | 0.000115 |
| 19 | CRI | 0.510638 | SLV | 0.501767 | | | | |
| 20 | SLV | 0.501742 | CIV | 0.459547 | | | | |
| 21 | CMR | 0.473856 | MEX | 0.434251 | | | | |
| 22 | CIV | 0.463023 | | | | | | |

Betweenness centrality measures the role of a country as a "bridge" between other trading partners. If a country with a high centrality value falls out of the network, it can considerably disrupt the trade flows of the network. This value is highest for trading intermediaries, such as the USA, Germany and Belgium, and the highest-ranking producers (Brazil, India, Kenya and Vietnam) have values less than half of the leading countries, both in 2018 and 2020. The leaders have experienced a slight loss in their centrality values from 2018 to 2020, but the range of values contracted (0.000035–0.010617 in 2018 and 0.000115–0.008533 in 2020), indicating a more even centrality distribution among countries.

Comparing the ranking by closeness centrality and betweenness centrality, out of the top 15 countries, 12 are the same in 2018, and 11 are the same in 2020. Betweenness centrality is moderately correlated with closeness centrality r = 0.614 in 2018 and r = 0.667 in 2020).

The eigenvector centrality measures indicate the countries having the most powerful trading partners (Table 4). The highest values were possessed by Spain, France, Greece, Italy, Israel, United Kingdom, Netherlands, Russia, Australia and Japan in 2018. In 2020, the top ten countries include Denmark and the United Arab Emirates, while Israel and Australia slipped back in the ranking. It is worth noting that African countries are not among the top-ranking countries, except for South Africa (ZAF, 13th in 2018), and Egypt was 40th in 2018. In 2020 the first African country is still South Africa, but its position is only 26th, followed by Egypt at 43rd, and no other African country in the first 50. Another interesting feature is that no major coffee producers are found among the top countries regarding eigenvector centrality, i.e., there are not many influential trade actors among their immediate connections. In 2018 the best-ranking coffee producers were India (39th), Indonesia (42nd) and Brazil (81st), while in 2020, Vietnam (36th), Indonesia (49th) and Brazil (67th).

**Table 4.** Eigenvector centrality measures value higher than 0.2 (2018, 2020 normalised).

| Eigenvector Centrality 2018 | | | | | | | | | Eigenvector Centrality 2020 | | | | | | | | |
|---|---|---|---|---|---|---|---|---|---|---|---|---|---|---|---|---|---|
| 1 | ESP | 1.0000 | 31 | POL | 0.82675 | 61 | PHL | 0.387445 | 1 | ESP | 1.0000 | 31 | IRL | 0.764102 | 61 | ARG | 0.386129 |
| 2 | FRA | 1.0000 | 32 | FIN | 0.759493 | 62 | GEO | 0.384447 | 2 | JPN | 0.98405 | 32 | LVA | 0.76033 | 62 | BHR | 0.377746 |
| 3 | GRC | 1.0000 | 33 | CHN | 0.758722 | 63 | OMN | 0.38329 | 3 | NLD | 0.98405 | 33 | PRT | 0.751476 | 63 | LBN | 0.377523 |
| 4 | ITA | 1.0000 | 34 | NZL | 0.757364 | 64 | DOM | 0.376823 | 4 | RUS | 0.98405 | 34 | QAT | 0.740375 | 64 | ISL | 0.374632 |
| 5 | ISR | 0.995576 | 35 | ROU | 0.751984 | 65 | BLR | 0.369642 | 5 | DNK | 0.956325 | 35 | SVN | 0.733736 | 65 | CYP | 0.3711 |
| 6 | GBR | 0.983449 | 36 | EST | 0.740312 | 66 | DZA | 0.367965 | 6 | ARE | 0.926986 | 36 | VNM | 0.696474 | 66 | CHL | 0.356011 |
| 7 | NLD | 0.983449 | 37 | BGR | 0.730474 | 67 | TUN | 0.350229 | 7 | FRA | 0.926986 | 37 | KWT | 0.688554 | 67 | MNG | 0.352181 |
| 8 | RUS | 0.983449 | 38 | MYS | 0.720865 | 68 | ARM | 0.316881 | 8 | GBR | 0.926986 | 38 | JOR | 0.687444 | 68 | BRA | 0.334417 |
| 9 | AUS | 0.98209 | 39 | IND | 0.719061 | 69 | ISL | 0.295639 | 9 | GRC | 0.926986 | 39 | NZL | 0.679858 | 69 | MLT | 0.326402 |
| 10 | JPN | 0.98209 | 40 | EGY | 0.715373 | 70 | ALB | 0.294439 | 10 | ITA | 0.926986 | 40 | OMN | 0.63874 | 70 | PAK | 0.323198 |
| 11 | SWE | 0.98209 | 41 | IRN | 0.704575 | 71 | PAN | 0.274449 | 11 | AUS | 0.911036 | 41 | ROU | 0.637394 | 71 | LUX | 0.308465 |
| 12 | TWN | 0.98209 | 42 | IDN | 0.69666 | 72 | MNG | 0.272789 | 12 | KOR | 0.911036 | 42 | POL | 0.624676 | 72 | TUN | 0.292269 |
| 13 | ZAF | 0.98209 | 43 | IRL | 0.685022 | 73 | SRB | 0.272789 | 13 | SGP | 0.911036 | 43 | EGY | 0.624252 | 73 | SYR | 0.283952 |
| 14 | CHE | 0.977666 | 44 | MEX | 0.650614 | 74 | SVK | 0.272789 | 14 | TWN | 0.911036 | 44 | CZE | 0.62105 | 74 | ECU | 0.281039 |
| 15 | PRT | 0.976191 | 45 | CZE | 0.643059 | 75 | LUX | 0.260124 | 15 | SAU | 0.909669 | 45 | LTU | 0.620034 | 75 | SVK | 0.277709 |
| 16 | ARE | 0.948056 | 46 | MAR | 0.631498 | 76 | HUN | 0.257444 | 16 | HKG | 0.902457 | 46 | IDN | 0.619652 | 76 | DOM | 0.274227 |
| 17 | UKR | 0.944642 | 47 | VNM | 0.608875 | 77 | MAC | 0.251096 | 17 | MYS | 0.894207 | 47 | EST | 0.619128 | 77 | VEN | 0.26747 |
| 18 | SGP | 0.892711 | 48 | KWT | 0.554221 | 78 | BGD | 0.242129 | 18 | DEU | 0.878997 | 48 | BGR | 0.601255 | 78 | KAZ | 0.259945 |
| 19 | LVA | 0.891353 | 49 | QAT | 0.520604 | 79 | BIH | 0.240739 | 19 | CAN | 0.871784 | 49 | IND | 0.543056 | 79 | MAC | 0.256252 |
| 20 | TUR | 0.888287 | 50 | HRV | 0.514302 | 80 | CRI | 0.240415 | 20 | CHN | 0.865138 | 50 | THA | 0.526531 | 80 | LBY | 0.255376 |
| 21 | HKG | 0.88252 | 51 | AUT | 0.510087 | 81 | BRA | 0.238609 | 21 | USA | 0.865138 | 51 | MEX | 0.521701 | 81 | BLR | 0.247851 |
| 22 | BEL | 0.879631 | 52 | LTU | 0.499857 | 82 | JAM | 0.231165 | 22 | UKR | 0.847524 | 52 | MAR | 0.505291 | 82 | ALB | 0.247472 |
| 23 | DEU | 0.879631 | 53 | JOR | 0.486933 | 83 | ECU | 0.226746 | 23 | BEL | 0.81472 | 53 | IRN | 0.484466 | 83 | KEN | 0.247236 |
| 24 | NOR | 0.871968 | 54 | THA | 0.462091 | 84 | MDV | 0.217222 | 24 | FIN | 0.792124 | 54 | PHL | 0.468864 | 84 | CUB | 0.243848 |
| 25 | USA | 0.865209 | 55 | LBN | 0.457385 | 85 | PAK | 0.210078 | 25 | SWE | 0.792124 | 55 | HRV | 0.464234 | 85 | HUN | 0.240481 |
| 26 | KOR | 0.863079 | 56 | ARG | 0.436081 | | | | 26 | ZAF | 0.792124 | 56 | GEO | 0.454382 | 86 | SRB | 0.240481 |
| 27 | CAN | 0.861721 | 57 | CHL | 0.421794 | | | | 27 | NOR | 0.784912 | 57 | DZA | 0.451202 | 87 | UZB | 0.239916 |
| 28 | SAU | 0.860362 | 58 | CYP | 0.396899 | | | | 28 | CHE | 0.779611 | 58 | ARM | 0.43482 | 88 | FRO | 0.233269 |
| 29 | DNK | 0.859426 | 59 | SYR | 0.395865 | | | | 29 | ISR | 0.772286 | 59 | AUT | 0.426223 | 89 | MNE | 0.231901 |
| 30 | SVN | 0.844187 | 60 | BHR | 0.393411 | | | | 30 | TUR | 0.764399 | 60 | URY | 0.406238 | 90 | SEN | 0.229364 |
| | | | | | | | | | | | | | | | 91 | LKA | 0.205484 |

### 3.3. Close Trading Communities as Modularity Classes

Modularity analysis was performed to identify country groups having closer trade connections within the group than with the outside world. Such trading communities are usually organised around one or a few exporters [2]. Modularity classes were defined by the weights as trade values (as % of the total), i.e., the strength of a connection was measured by the percentage trade value from the source to the target country. The modularity index measures the quality of community division, comparing it to a similar quality random network. The modularity index for 2018 was 0.270 in 2018 and 0.254 in 2020. As modularity should theoretically fall between −1 and +1, these values indicate a moderately strong linkage within the communities, though the strength is somewhat less in 2020 than in 2018.

Similar to [2], our analysis identified 5–5 major trading communities in 2018 and 2020. However, these communities differed considerably.

In 2018 the largest community had Brazil, Uganda and India as major producers, with Germany and many European and Asian countries. The second group had Columbia, Guatemala and Peru as leading exporters and the USA, Canada and Australia as main importers. The third community included exporters Mexico, Indonesia and Vietnam, with importers Britain, Japan, Russia and many European countries. The fourth group had Costa Rica and Honduras as the main producers, with Belgium, France and the Netherlands as the main importers. The fifth group had Ethiopia as the main exporter, with some African and Arabic countries as importers. This structure is rather similar to that of Utrilla-Catalan et al. [2] for the year 2017 and before, especially the last two groups concentrated around Belgium and around Ethiopia. However, in 2017, the 2018 communities of Brazil and Colombia were still merged, and Indonesia and Vietnam formed separate communities.

In 2020, out of the 235 territories, the largest five communities contained 151 countries, i.e., 64.26% of all countries altogether. The largest four communities contain one or two major producers: the first one has Brazil, the second one has India, Indonesia, Uganda and Ethiopia, the third one has Vietnam, and the fourth one has Colombia. The fifth community is organised around an importer, Belgium, with only one coffee exporter, Costa Rica, in the network (Table 5, Figure 4). The rest of the countries (more than 35% of all countries) do not belong to any of the five large communities, i.e., they do not have strong connections to any group of countries.

**Table 5.** Trading communities in 2018 and in 2020 (countries not listed do not belong to any of the strongest 5 classes).

| | C1—2018 | | C2—2018 | | C3—2018 | | C4—2018 | C5—2018 | C1—2020 | | C2—2020 | | C3—2020 | | C4—2020 | C5—2020 |
|---|---|---|---|---|---|---|---|---|---|---|---|---|---|---|---|---|
| 1 | AGO | LBN | ABW | SLV | ARM | THA | BEL | CUW | ARG | SLV | ALB | SGP | AUT | SEN | ABV | BEL |
| 2 | ALB | LBY | AUS | SOM | CHN | TLS | BIH | ERI | BHR | SVK | ARE | SSD | AZE | SRB | AFG | BIH |
| 3 | ARE | LKA | BGD | SSD | CIV | TUN | CMR | ETH | BLZ | SVN | ARM | SWZ | BGR | THA | AUS | CRI |
| 4 | ARG | LTU | BHS | TCA | CUB | TZA | COD | QAT | BRA | SWE | BGD | TWN | BLR | UKR | BHS | FRA |
| 5 | ATF | LVA | BLZ | TWN | DZA | VEN | CRI | SAU | CHE | SYC | BTN | UGA | BRN | UZB | BMU | LUX |
| 6 | AUT | MDA | BMU | USA | ECU | VNM | FRA | SDN | CHL | SYR | COD | UMI | CIV | VNM | BRB | NLD |
| 7 | AZE | MDG | BRB | XXY | EGY | ZAF | HND | SWZ | CHN | TTO | DJI | YEM | CZE | XXX | CAN | |
| 8 | BEN | MDV | CAN | | ESP | | HTI | YEM | CPV | TUN | EGY | ZMB | DEU | | CMR | |
| 9 | BGR | MKD | CHE | | GBR | | LUX | | CUB | TUR | ETH | | DZA | | COL | |
| 10 | BHR | MNE | COL | | GEO | | MLT | | CYP | TZA | GEO | | ECU | | CUW | |
| 11 | BLR | MNG | CYM | | HKG | | NLD | | DNK | URY | GHA | | ESP | | CYM | |
| 12 | BRA | NCL | DJI | | IDN | | PRK | | DOM | VEN | HKG | | FRO | | GRL | |
| 13 | BTN | NFK | DMA | | IRN | | SYC | | EST | ZAF | HTI | | HND | | ISL | |
| 14 | CHL | NPL | DOM | | JAM | | XXX | | FIN | | IDN | | HUN | | JAM | |
| 15 | CPV | OMN | GTM | | JPN | | | | GBR | | IND | | IRL | | KEN | |
| 16 | CYP | POL | IRL | | KHM | | | | GRC | | IRN | | KAZ | | KOR | |
| 17 | CZE | PRY | ISR | | LAO | | | | GTM | | IRQ | | KHM | | LCA | |
| 18 | DEU | SRB | KEN | | MAR | | | | HRV | | ISR | | LKA | | MAC | |
| 19 | DNK | SVK | KOR | | MEX | | | | ITA | | JOR | | LTU | | MHL | |
| 20 | EST | SVN | LCA | | MMR | | | | JPN | | KWT | | MDA | | NIC | |
| 21 | FIN | SWE | MAC | | MUS | | | | LBN | | LBY | | MDV | | NOR | |
| 22 | GRC | SYR | MHL | | MYS | | | | LBR | | MAR | | MLT | | NZL | |
| 23 | HRV | TJK | NGA | | PAK | | | | LVA | | MKD | | MMR | | PER | |
| 24 | HUN | TTO | NIC | | PHL | | | | MEX | | MYS | | MNG | | ROU | |
| 25 | IND | TUR | NOR | | PNG | | | | MNE | | NGA | | MUS | | SOM | |
| 26 | ISL | UGA | NZL | | PRT | | | | MOZ | | NPL | | PAN | | TCA | |
| 27 | ITA | UKR | PAN | | PSE | | | | PAK | | OMN | | PHL | | USA | |
| 28 | JOR | URY | PER | | RUS | | | | PRY | | QAT | | POL | | | |
| 29 | KAZ | | PYF | | SEN | | | | PSE | | SAU | | PRK | | | |
| 30 | KWT | | ROU | | SGP | | | | RUS | | SDN | | PRT | | | |
| No. of countries | 58 | | 37 | | 37 | | 14 | 8 | 43 | | 38 | | 37 | | 27 | 6 |
| % of countries | 24.8% | | 15.8% | | 15.8% | | 5.98% | 3.42% | 18.3% | | 16.2% | | 15.7% | | 11.5% | 2.6% |

Comparing the two years, the community around Belgium lost one of its producers (Honduras), which merged with the community around Vietnam, while Ethiopia lost its entire trading community, merging with the community organised around India, Indonesia and Uganda. The community around Colombia and the Central-American exporters remained nearly unchanged. The trading partners in the community around Brazil considerably changed, as Germany moved out from this group and joined the group of Vietnam, while the former importers belonging to the group of Vietnam moved to the group led by Brazil. This indicates that the trade structure changed considerably around the major producers. Generally, from the members of the five communities in 2018, about half of the countries remained together in 2020, and the other half dispersed among different communities, indicating a rather unstable structure of trading communities. For example, from the 58 countries of C1 in 2018, only 22 remained together in C1 2020, 16 countries moved to C3, 11 to C2, one country to C4 and eight countries did not join any community in 2020, while 21 countries moved into C1 from other communities.

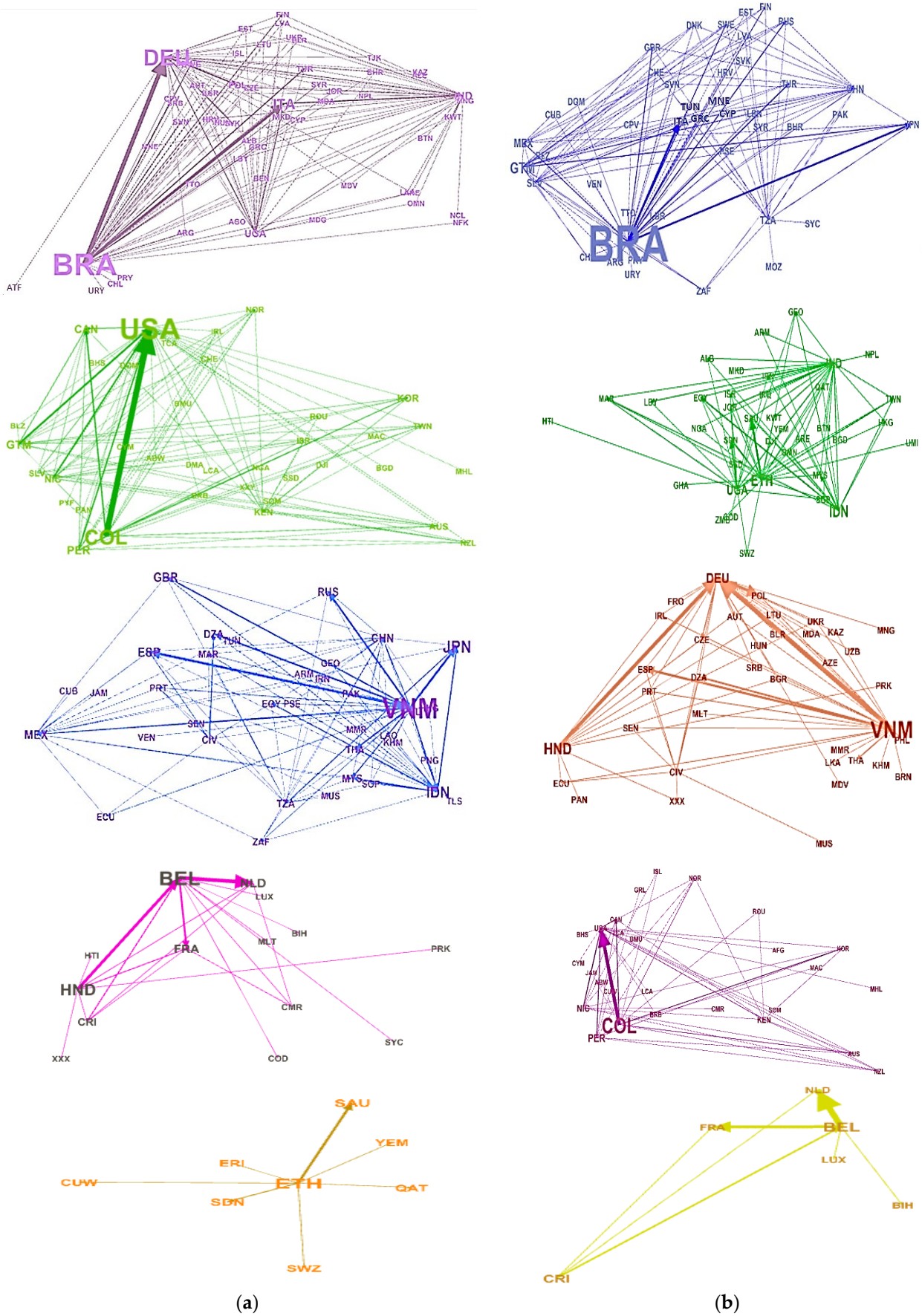

**Figure 4.** Trading communities in 2018 and 2020: (**a**) Year 2018; (**b**) Year 2020.

The total trade value within the communities gave 45.8% of the total global trade in 2018 and 39.1% in 2020. This means that the communities are rather concentrated, although, in 2020, trade ties loosened somewhat.

Another interesting feature of network structure is the minimum spanning tree, in which the sum of the edge weights is minimised in the connected network. For a trade network, however, the important feature is the connections with maximum trade values. Therefore, for us, a "maximum weight" spanning tree would be more useful. In order to perform this analysis, the weights of the edges were transformed into a metric that is small when export is high, and vice versa. For this purpose, the method introduced by Mantegna [43] and applied by many other authors (e.g., see [31]). In order to create edge weights that comply with the requirements of a distance, each trade value is converted to a metric distance as follows [43]: d(i,j) = square root[2 × (1 − w(i,j)/W)], where d(i,j) is the "distance" between nodes i and j, w(i,j) is the trade value from i to j, and W is the total sum of the trade values in the network. Small values of d(i,j) imply large trade values between the two countries, and MST is constructed with these distance values as edge weights (Figure 5, Table 6). This means that the minimum spanning tree includes the trade linkages having the highest values, i.e., the most important trade partners for each country.

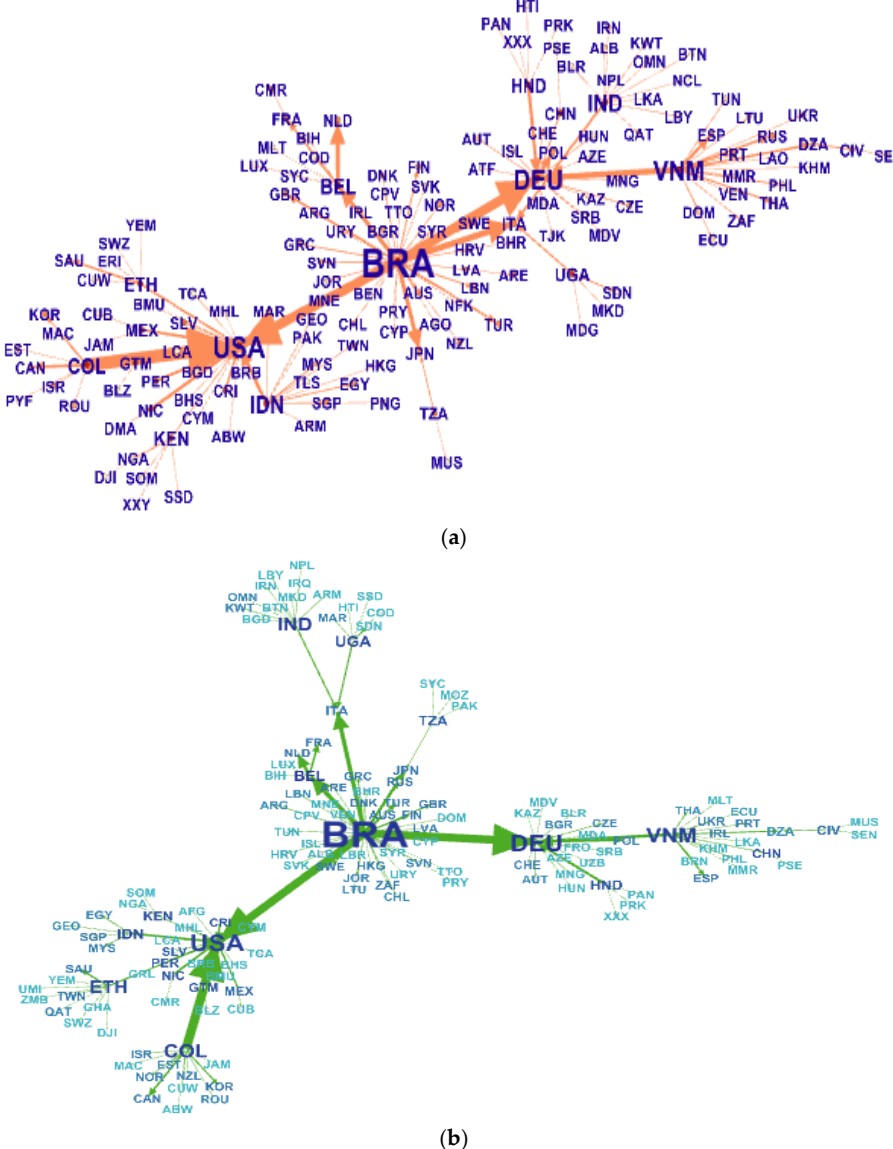

(**a**)

(**b**)

**Figure 5.** The minimum spanning tree, with node size proportional to node degree (in + out), edge thickness representing trade value as weight. (**a**) Year 2018; (**b**) Year 2020.

**Table 6.** The source and target countries in the MST, 2018 and 2020.

| 2018 Source (22) | MST Edges % of Netw | MST Edge No. | Netw Edge No. | MST Value % of Netw | Country Share in MST Value | Country Share in Netw Value | 2020 Source (21) | MST Edges % of Netw | MST Edge No. | Netw Edge No. | MST Value % of Netw | Country Share in MST Value | Country Share in Netw Value |
|---|---|---|---|---|---|---|---|---|---|---|---|---|---|
| BRA | 40.9% | 36 | 88 | 81.0% | 37.592% | 24.7% | BRA | 41.9% | 39 | 93 | 81.6% | 43.646% | 29.2% |
| COL | 11.4% | 08 | 70 | 57.4% | 13.857% | 12.9% | COL | 16.0% | 12 | 75 | 55.1% | 14.482% | 14.4% |
| VNM | 26.6% | 17 | 64 | 47.2% | 12.616% | 14.3% | VNM | 23.1% | 15 | 65 | 38.2% | 7.756% | 11.1% |
| BEL | 12.5% | 7 | 56 | 81.6% | 5.001% | 3.3% | BEL | 6.0% | 4 | 67 | 79.8% | 5.330% | 3.6% |
| IDN | 20.7% | 12 | 58 | 57.7% | 4.966% | 4.6% | IDN | 8.6% | 5 | 58 | 42.4% | 3.689% | 4.7% |
| DEU | 20.3% | 14 | 69 | 53.5% | 3.150% | 3.1% | DEU | 20.0% | 15 | 75 | 53.4% | 3.140% | 3.2% |
| GTM | 3.7% | 2 | 54 | 41.4% | 2.996% | 3.9% | ETH | 14.7% | 10 | 68 | 35.4% | 3.018% | 4.7% |
| HND | 8.8% | 5 | 57 | 23.0% | 2.726% | 6.3% | HND | 7.3% | 4 | 55 | 29.2% | 2.727% | 5.1% |
| ETH | 8.6% | 6 | 70 | 31.0% | 2.618% | 4.5% | GTM | 3.8% | 2 | 52 | 38.5% | 2.690% | 3.8% |
| NIC | 4.0% | 2 | 50 | 53.2% | 2.370% | 2.4% | UGA | 9.5% | 6 | 63 | 44.3% | 2.448% | 3.0% |
| IND | 17.1% | 12 | 70 | 37.5% | 2.045% | 2.9% | NIC | 3.7% | 2 | 54 | 47.5% | 2.237% | 2.6% |
| MEX | 7.7% | 3 | 39 | 59.5% | 1.981% | 1.8% | PER | 4.3% | 2 | 46 | 27.0% | 1.859% | 3.8% |
| PER | 2.1% | 1 | 48 | 26.8% | 1.904% | 3.8% | CRI | 2.1% | 1 | 47 | 49.0% | 1.716% | 1.9% |
| CRI | 2.4% | 1 | 42 | 48.1% | 1.618% | 1.8% | IND | 15.3% | 11 | 72 | 34.5% | 1.712% | 2.7% |
| UGA | 6.3% | 4 | 63 | 33.6% | 1.558% | 2.5% | MEX | 18.2% | 2 | 11 | 62.4% | 1.653% | 1.4% |
| CIV | 10.5% | 2 | 19 | 51.3% | 0.753% | 0.8% | CIV | 14.3% | 3 | 21 | 61.5% | 0.596% | 0.5% |
| CHN | 3.9% | 2 | 51 | 30.0% | 0.613% | 1.1% | KEN | 6.0% | 3 | 50 | 21.9% | 0.491% | 1.2% |
| TZA | 5.0% | 2 | 40 | 35.1% | 0.542% | 0.8% | SLV | 2.6% | 1 | 38 | 38.1% | 0.442% | 0.6% |
| SLV | 2.9% | 1 | 35 | 43.1% | 0.519% | 0.6% | TZA | 8.9% | 4 | 45 | 23.4% | 0.361% | 0.8% |
| KEN | 10.7% | 6 | 56 | 17.8% | 0.432% | 1.3% | CHN | 2.3% | 1 | 44 | 0.5% | 0.006% | 0.7% |
| CMR | 4.8% | 1 | 21 | 36.5% | 0.130% | 0.2% | USA | 13.1% | 8 | 61 | 0.1% | 0.002% | 0.9% |
| USA | 12.9% | 9 | 70 | 0.8% | 0.012% | 0.9% | | | | | | | |
| SUM | 12.9% | 153 | 1190 | 53.3% | 100% | 100% | TOTAL | 12.9% | 150 | 1160 | 54.6% | 100% | 100% |

| Target Area (135) | MST edges % of Netw | MST edges (153) | Network edges | MST import value % of Netw | Country share in MST value | Country share in Netw value | Target Area (134) | MST edges % of Netw | MST edges (153) | Network edges | MST import value % of Netw | Country share in MST value | Country share in Netw value |
|---|---|---|---|---|---|---|---|---|---|---|---|---|---|
| USA | 55.0% | 11 | 20 | 86.1% | 34.723% | 22.5% | USA | 57.9% | 11 | 19 | 87.6% | 35.006% | 22.3% |
| DEU | 19.0% | 4 | 21 | 62.9% | 15.850% | 14.0% | DEU | 15.0% | 3 | 20 | 62.2% | 16.484% | 14.8% |
| ITA | 13.6% | 3 | 22 | 53.7% | 7.513% | 7.8% | ITA | 15.0% | 3 | 20 | 54.3% | 7.027% | 7.2% |
| JPN | 10.0% | 2 | 20 | 35.2% | 3.979% | 6.3% | BEL | 5.3% | 1 | 19 | 35.3% | 5.026% | 8.0% |
| NLD | 4.8% | 1 | 21 | 58.1% | 3.585% | 3.4% | JPN | 10.0% | 2 | 20 | 33.7% | 3.496% | 5.8% |
| BEL | 4.8% | 1 | 21 | 23.4% | 3.237% | 7.7% | NLD | 5.0% | 1 | 20 | 59.8% | 3.414% | 3.2% |
| POL | 5.6% | 1 | 18 | 76.6% | 2.231% | 1.6% | POL | 6.7% | 1 | 15 | 74.1% | 2.093% | 1.6% |
| DZA | 20.0% | 2 | 10 | 90.0% | 2.106% | 1.3% | CAN | 5.3% | 1 | 19 | 29.9% | 1.944% | 3.6% |
| ESP | 4.5% | 1 | 22 | 39.9% | 2.084% | 2.9% | FRA | 5.0% | 1 | 20 | 31.7% | 1.884% | 3.3% |
| CAN | 5.3% | 1 | 19 | 30.2% | 1.918% | 3.5% | TUR | 5.9% | 1 | 17 | 90.3% | 1.639% | 1.0% |
| RUS | 4.8% | 1 | 21 | 50.5% | 1.543% | 1.7% | DZA | 22.2% | 2 | 9 | 81.3% | 1.583% | 1.1% |
| FRA | 9.1% | 2 | 22 | 26.8% | 1.518% | 3.2% | SAU | 5.6% | 1 | 18 | 74.3% | 1.471% | 1.1% |
| GBR | 4.8% | 1 | 21 | 29.8% | 1.471% | 2.7% | ESP | 4.8% | 1 | 21 | 27.3% | 1.342% | 2.8% |
| TUR | 5.6% | 1 | 18 | 91.6% | 1.421% | 0.9% | RUS | 5.0% | 1 | 20 | 34.6% | 1.197% | 1.9% |
| SAU | 5.6% | 1 | 18 | 78.6% | 1.253% | 0.9% | KOR | 5.3% | 1 | 19 | 25.4% | 1.186% | 2.6% |
| THA | 16.7% | 1 | 6 | 97.2% | 1.037% | 0.6% | GBR | 5.0% | 1 | 20 | 27.3% | 1.033% | 2.1% |
| SWE | 5.0% | 1 | 20 | 34.5% | 0.980% | 1.6% | SWE | 5.6% | 1 | 18 | 37.7% | 1.011% | 1.5% |
| KOR | 5.0% | 1 | 20 | 22.0% | 0.890% | 2.3% | SDN | 33.3% | 1 | 3 | 76.5% | 0.943% | 0.7% |
| FIN | 5.6% | 1 | 18 | 44.5% | 0.817% | 1.0% | FIN | 5.6% | 1 | 18 | 38.2% | 0.711% | 1.0% |
| MYS | 5.9% | 1 | 17 | 37.2% | 0.708% | 1.1% | GRC | 5.0% | 1 | 20 | 52.9% | 0.684% | 0.7% |
| first 20 area | 10.1% | 38 | 375 | 49.5% | 88.86% | 86.99% | | 10.1% | 36 | 355 | 57.65% / 49.8% | 89.2% | 86.4% |
| rest MST | 16.0% | 115 | 721 | 6.2% | 11.136% | 13.0% | | 15.7% | 114 | 727 | 44.55% / 6.0% | 10.8% | 13.6% |

The minimum spanning tree in 2018 contains 153 edges, and its total value is 7.3701 representing 53.3% of the total trade in the network. In 2020 the MST contained 150 edges, and its total value was 6.3732, referring to 54.6% of the total network trade.

The major hubs in the MST were Brazil and Vietnam (exporters) and the USA and Germany (importers) in 2018. The structure of the MST is similar in 2020, but some secondary hubs (around Ethiopia, Colombia, India, Uganda and Italy) seem to emerge. An

important change between the two years is the decrease in the average weighted degree of the two years. The MST contains all countries involved in the coffee trade. Brazil, Colombia and Vietnam have outstanding roles as source countries. In 2018 they represented 64%, and in 2020 nearly 66% of the trade in the MST. However, Vietnam's weight considerably decreased in 2020, while Brazil and Colombia increased their share. The importer side is more heterogeneous. Table 6 lists the top 20 countries of the importer side, covering 49.5–49.8% of the MST trade value. The list of target countries is led by the USA, Germany and Italy, and the two years are rather similar again.

Table 7 summarises the overall measures of the network, comparing the parameters of the modularity classes and the MST to that of the full network. As the table illustrates, the modularity classes reflect a more compact, tighter structure than the whole network. Comparing the two years the full network contracted in the COVID-year of 2020, both in the number of trade connections (smaller average degree), trade values (smaller average weighted degree and graph density) and more disrupted overall structure (larger network diameter). The trading communities became generally smaller, and in 2020, none of them reached the size of the two largest communities in 2018 by their share of the total trade value.

**Table 7.** Summary of network parameters for 2018 and 2020.

| 2018 | Full Network | C1 | C2 | C3 | C4 | C5 | MST |
|---|---|---|---|---|---|---|---|
| Average degree | 5.085 | 2.397 | 2.649 | 2.784 | 1.5 | 0.875 | 0.645 |
| Avg weighted degree | 75316.329 | 49291.3 | 70493.4 | 51447.4 | 69460.4 | 19139.0 | 0.923 * |
| as % of total graph or cluster trade | 0.4274% | 1.7391% | 2.7021% | 3.4486% | 7.1429% | 12.5000% | 0.0000% |
| Network diameter | 4 | 2 | 2 | 3 | 2 | 1 | 2 |
| Graph density | 0.022 | 0.042 | 0.074 | 0.077 | 0.115 | 0.125 | 0.003 |
| Avg. path length | 1.781 | 1.280 | 1.44 | 1.658 | 1.429 | 1.00 | 1.514 |
| Total trade value | 17624021 | 2834230 | 2608822 | 1491840 | 972446 | 153112 | 9390046 |
| % of total graph trade | 100% | 16.1% | 14.8% | 8.5% | 5.5% | 0.9% | 53.3% |
| Modularity | 0.270 | | | | | | 0.807 |
| 2020 | Full Network | C1 | C2 | C3 | C4 | C5 | MST |
| Average degree | 4.936 | 2.535 | 2.158 | 1.730 | 1.963 | 1.167 | 0.641 |
| Avg weighted degree | 72519.2 | 46915.6 | 18184.7 | 36474.0 | 76727.5 | 94578.7 | 0.901 * |
| as % of total graph or cluster trade | 0.4255% | 2.3256% | 2.6359% | 2.7557% | 3.7033% | 16.6667% | 0.0000% |
| Network diameter | 5 | 2 | 2 | 3 | 2 | 2 | 2 |
| Graph density | 0.021 | 0.06 | 0.058 | 0.048 | 0.075 | 0.233 | 0.003 |
| Avg. path length | 1.784 | 1.180 | 1.261 | 1.559 | 1.459 | 1.222 | 1.479 |
| Total trade value | 17042001 | 2017373 | 689879 | 1323568 | 2071879 | 567472 | 9300152 |
| % of total graph trade | 100% | 11.8% | 4.0% | 7.8% | 12.2% | 3.3% | 54.6% |
| Modularity | 0.254 | | | | | | 0.796 |

* The MST weights are the metric-transformed values of exports.

## 4. Discussion

Our aim was to analyse the change in the global coffee green bean trade network due to the COVID-19 pandemic. For this purpose, the pre-COVID year 2018 and the year 2020 were compared. Our first question was how the major actors changed between 2018 and 2020. The in- and out-degree rankings of the countries in these two years listed nearly the same countries, i.e., the exporters having the largest number of trade partners did not change from 2018 to 2020, nor did the majority of the importers with the most trade partners.

Our results show that from 2018 to 2020, while importers lost some trade partners, the number of trade partners actually increased for the top 10 exporter countries, indicating a more diversified trade structure for most of them.

The distribution of trade values shows a somewhat different picture according to the weighted in- and out-degrees. The dominant exporters are Brazil, Vietnam and Colombia in both 2018 and 2020, while the major importers are the USA, Germany, Belgium, Italy, Japan, Canada, the Netherlands and France both in 2018 and 2020. However, although the major actors remained the same, some shifts took place among them. By 2020 Brazil and Germany increased the number of trade connections and trade share, while the USA, a major consumer and intermediary country, actually lost partners, though it managed to maintain its trade share. Vietnam and Honduras suffered considerable losses in their share of international trade, even though Vietnam could actually increase the number of trade partners, trying to diversify its trade linkages. According to the findings in [2] for green bean coffee between 1995–2018, the large exporters increased their trade values and trade connections up to 2018. Our results for 2020 show that from 2018 to 2020, approximately half of the exporters could increase the number of their partners and trade value, which indicate that only half of the countries could cope with the challenges of the pandemic. The absolute winner among exporters is Brazil, with considerably more trade relationships and a larger share of global trade in 2020 than in 2018. Smaller gains and smaller losses were experienced by other exporters. Among major importers, no such substantial gains or losses could be identified. Thus we may conclude that the list of the largest actors remained the same, and Brazil is the only country than actually increased its importance considerably.

These findings give the answer to our second question: importers seem to suffer some losses in the number of their trade partners, while the major exporters somewhat widened their trade contacts. However, the distribution of the trade shares changed more for importers and for exporters, too. This is in line with the findings of [55] stating that demand for coffee did not suffer significant losses during the pandemic. Therefore, former trade partnerships did not break up. Only a slight decrease occurred.

The findings in [2] state that up to 2018, the green coffee trade has changed its structure. Originally, the trade was focused on traditional coffee producers, but by 2017 the leaders were the largest coffee producers and some non-producing countries. These changes in the structure of the international green coffee market have led to greater inequality between producing and importing countries.

As the degree distribution in our research showed, Belgium, the USA, Germany and China are the only intermediary countries that actually import from nearly all major producers, having 19–21 import partnerships in 2018 and 19–20 partnerships in 2020. Therefore, their intermediary roles have remained strong after the COVID outbreak.

The correlation between the weighted degrees of 2018 and 2020 is very high (r = 0.9809), which means that the overall pattern of country shares in global trade did not change much from 2018 to 2020. The data clearly show that the core actors are the same in both years (importers: USA, Germany, Belgium and Italy; exporters: Brazil, Colombia and Vietnam).

The third research question was to see possible changes in the closer trading communities. By modularity analysis, country groups were identified with closer trade connections within themselves than with the outside world. Both in 2018 and 2020, 5–5 such major trading communities were identified, organised around one or a few major exporters, which is similar to the findings of [2] about coffee trade and of [25] regarding aggregate net exports. Our clusters demonstrate more intensive within-community trade than that between communities. The cluster densities are all higher than that of the full network, meaning that there are considerably more trade partnerships within the trading communities than across them. This idea is also supported by the higher average weighted degrees and the smaller network diameter and average path length, too.

However, our analysis showed that the communities considerably differed between 2018 and 2020. In 2018 the largest community had Brazil, Uganda and India as major producers, with Germany and many European and Asian countries. The other trade

communities were organised around Columbia, Indonesia and Vietnam; Costa Rica and Belgium, and Ethiopia. This structure is rather similar to that of [2] for the year 2017, especially the last two groups concentrated around Belgium and Ethiopia, respectively. By 2020, the largest five communities were concentrated around: Brazil; India–Indonesia– Uganda–Ethiopia; Vietnam; Colombia; Belgium–Costa Rica. Comparing the two years, the community around Belgium shrank, with one exporter partner (Honduras) merging with the Vietnam-led community. Ethiopia completely lost its leading role in its former community and merged with the community around India. This community also absorbed Indonesia and Uganda, formerly attached to Brazil. The community around Colombia and the Central-American exporters remained nearly unchanged, while the community around Brazil considerably changed, as its major importers changed place with those formerly attached to Vietnam.

Further analysis is needed to explain the reasons for these particular changes, but the average path lengths may indicate the importance of more direct trade linkages initiated by the pandemic. The number of partners among trading communities became more evenly distributed by 2020, with only one small community sticking together. The same is true for the share of total trade within particular communities. Similar changes were found in [25] about net exports, establishing more fragmented clusters in 2020 than in 2019, attributing the change to the impacts of the pandemic. The results in [29] also underline the instability of modularity-based trading clusters analysing a longer time period.

The fourth of our research questions focused on the core–periphery situation in the coffee trade network. The closeness and the betweenness centrality index values can be used to give meaningful insights. The countries with the highest betweenness centrality have a core role in the network, being able to control trade flows. Not surprisingly, the largest such between-centrality values belong to the large intermediaries: the USA, Germany, Belgium and China, together with the main producers: Brazil, Indonesia, India, Vietnam, Colombia and Uganda. Their roles did not show much change from 2018 to 2020. Locally important countries with strong core roles within their immediate neighbourhoods are those with high closeness centralities. Brazil is the strongest among them, followed by Germany, Ethiopia, Colombia and the USA. In this list, the change between 2018 and 2020 is more striking: in particular, the role of the USA considerably weakened, while other countries suffered only a minor loss of centrality.

Due to COVID-19, the difference between countries increased, and the USA and Mexico suffered a considerable decrease in their closeness centrality value—meaning a loss in their regional importance. The high eigenvector centralities characterise countries with the most powerful trading partners. These were Spain, France, Greece, Italy, United Kingdom, Netherlands, Russia and Japan in both 2018 and 2020. However, in 2018 Israel and Australia were also among the top ten. By 2020 they had lost their position against Denmark and the United Arab Emirates. Neither African countries nor major coffee producers are found among the top countries.

From 2018 to 2020, the betweenness centrality ranking of countries changed only slightly, but the closeness centrality ranking changed more—reflecting that only a part of the countries could maintain its significance in their neighbourhoods. This is in agreement with the results in [51] for overall trade and trade in various product groups.

It is also worth noting that betweenness and closeness centralities moderately correlate both in 2018 and 2020, indicating that the countries acting as global "bridges" only partly coincide with those being important locally. This is the same as the findings in [25] regarding net export values or the results in [26] regarding trade in plastic scrap.

The core–periphery relationship is reflected by the minimum spanning trees, identifying the backbone of the global trade network. The share of the trade value of the MST within the network slightly increased from 2018 to 2020. The disruption of trade relationships resulting from the pandemic is reflected among exporters by the loss of the importance of Vietnam and Mexico, the increased importance of Brazil, Colombia, Ethiopia and Uganda, and the considerable loss of imports by former leading countries Japan, the

Netherlands, Britain and Spain, with the striking rise of Belgium. The countries able to maintain their core positions are Brazil, Colombia, Belgium and the USA, while Vietnam, Mexico, the Netherlands, Britain and Spain weakened their former core roles.

These changes support the assumption that the COVID-19 pandemic affected the coffee green bean trade network considerably—leading to losses of trade positions for many Asian and African countries and some of the major importer-intermediary countries. These results may be due to the formerly very heterogeneous and long-distance trade relations of these countries. Although some of the traders actually diversified their trade connections, increasing the number of trading partners, these attempts could not compensate for losses in trade values. Many of the negatively affected countries lost not only trade value but trading partners alike.

Similar findings were established about the impacts of the pandemic on aggregate trade. The year 2020 experienced some of the largest reductions in trade and output volumes since World War II, regarding both world industrial production and goods trade, although the trade impacts across specific goods and trade partners highly differ. The food and agricultural products were less impacted than minerals, vehicle manufacturing and the services sector [62]. The effects of the economic downturn on global trade have been fast and intensive, affected by not only the decline in global demand but by enhanced cross-border restrictions, port closures and other logistical disruptions. Overall, global trade declined by about $2.5 trillion in 2020, by about 9% compared with 2019 [63]. Trade in essential products such as foodstuffs was rather resilient, while trade in pharmaceuticals, medical devices and personal protective equipment and related goods actually increased in 2020, as well as in home office and fitness equipment. By mid-2021, the value of international trade was already substantially higher than pre-pandemic levels in all sectors except energy products [63].

Regarding the coffee trade, the extent to which a country was impacted by the pandemic is determined by many factors. Countries in which the coffee harvest coincided with the peak of the pandemic were more severely affected due to restricted labour supply, and coffee growers employing a large number of migrant workers were more likely to be affected than smallholders primarily relying on family [64]. The COVID-19 pandemic changed consumption patterns, i.e., the demand in the global coffee market, particularly as coffee is considered a social drink. Therefore, the lockdowns significantly decreased out-of-home coffee consumption, but home deliveries and increased home consumption compensated for it. Consumers shifted to traditional brands, causing a considerable loss for upcoming coffee brands. Altogether, coffee consumption did not change much during the COVID-19 pandemic situation, in spite of the restrictions and lockdowns, but consumption methods underwent changes [55].

Regarding the coffee trade, the impacts of the pandemic have not been analysed by network methodology, but former research [2] applied similar tools to the coffee trade between 1995 and 2018. Our findings of 2018 are similar to theirs, and contrasts of 2020 are presented not only to 2018 in our analysis but to the results of former years in their analysis. In [2], the major importers, according to the number of their trade partners, were the same as in our analysis: Belgium, the USA and Germany—but the UK, Japan and Denmark were also found important, though in our analysis their roles were somewhat weaker, while we found Spain and France to have the largest numbers of exporter partners. When trade volumes were considered, [2] identified USA, Germany and Japan together with Belgium, Italy and France. These results are completely in line with our findings for both years. Regarding exporters, [2] lists Vietnam, Brazil, Colombia, India, Indonesia, Honduras, Guatemala and Peru as the most important ones, and our results add Ethiopia to this list, as well as major intermediary countries, such as Belgium and Germany. Regarding the centrality results, [2] identifies a long-term tendency of decreasing closeness centrality of small coffee producers from 1995 to 2018, and our findings also agree with this for 2018 and 2020. The leading role in the closeness centrality of large producers and USA and European importers are found in [2] as well as in our results. Regarding betweenness centrality,

our findings for 2018 and 2020 are also in line with the results of [2] for 2017 and 2018, i.e., Germany and Belgium being at the top, with India. However, our results show that the USA has achieved an even stronger position than these in 2018 and in 2020. Regarding the formation of closely-knit trading communities, [2] found an evolution from the more evenly distributed structure of trading groups in 1995 towards a few large and a few small communities by 2017. Our findings support this tendency in 2018 and in 2020, too. A similar analysis of the wheat market was done in [27], and they found similar features of the global wheat market in 2018–2019 as the coffee market results in our study. The wheat market, with 94 countries and 1041 trade relationships, produced a graph density of 0.119, i.e., five times as dense a network as in the case of the coffee market—which is explained by the much smaller number of trading partners, and more countries being able to produce wheat for themselves. Coffee requires much more specific growing conditions and therefore has only 21–22 producers. However, it has worldwide demand, which necessarily leads to a less balanced trading structure. The average path length in the wheat network was found to be 2.397, which is about twice as high as our findings for the coffee market—which again shows that the coffee market is more characterised by direct links from a producer to an importer than the wheat network. The findings in [39] about wheat trade from 1992 to 2017 show a decreasing graph density from 0.1 to 0.07, with average path length decreasing from 2.5 to 2.8. The number of countries and the time span differed from [27], though the number of connections was limited to only 365 partnerships. These results of network analysis of agricultural commodities illustrate the suitability of network metrics for global commodity trade networks, though they did not assess the effects of the COVID-19 pandemic on network characteristics.

## 5. Conclusions

Our analysis compared the typical features of the international coffee bean trade network between the years 2018 and 2020. The analysis was performed using tools of social network analysis and revealed considerable changes between the pre-pandemic and the pandemic year. The role of the major actors changed considerably. Many trade relationships became disrupted, and as well as the decrease in overall trade, the number of trade connections also changed—with some countries gaining but more countries losing compared to their former positions. Similar patterns of changes have been experienced in other sectors of the economy, and the methods of social network analysis have been found useful for analysis.

Because of the interrelated character of the coffee industry with labour, farmers, firms, suppliers, traders, exporters, consumers and financial institutions, etc. involved, in case of a problem in any part of the supply chain, the result could be disruption either globally or within a country [65]. In recent years the unexpected problem of the COVID-19 pandemic represented an extraordinary joint supply and demand shock to the global coffee sector [66,67]. During COVID-19, the smallholders-dominated supply side encountered some constraints that significantly impacted coffee production, affecting productivity and marketing, and especially producers' probability of market participation [66]. The implementation of social distancing and regional restrictions impacted the availability of labour and other inputs. Farmers were less able to manage their farms, creating favourable conditions for Coffee Leaf Rust and other pests and diseases while being more vulnerable to drought or hurricanes [65], affecting the sustainability of coffee production. Declining household income, difficulties in covering farming costs, the related issues of the intensity of cultivation, particularly purchasing and fertilisation decisions [68], and decreasing investments in coffee plantations create conditions favourable for future shocks, which, in turn, are likely to drive the coffee industry into another severe production crisis [69]. The threats that COVID-19 pose to the global coffee sector is daunting with profound implications for coffee production [67]. Despite the problems and impending risks on the production side, global coffee consumption has steadily increased in the last decades [70]. The lockdowns induced by the pandemic, however, led to changing consumption patterns, including

increasing home deliveries and more drinks ordered via apps to reduce exposure [71]. The lockdown measures throughout the main coffee-consuming countries (i.e., Europe and North America) led to a drop in out-of-home sales, as most cafés and restaurants closed [70]. As a result of the pandemic, demand in both export and local markets has changed, having a financial impact on buyers, processors, retailers and consumers at home and abroad [72].

The global coffee industry suffered more from climate and weather-related issues than the pandemic. The example of two of the leading exporters illustrates the differences in impacts related to the development policy of the national coffee producer sector. As a strong market player, Brazil has years of experience and several advantages that help it cope with the impacts of the pandemic and remain globally competitive. Although the country had to face difficulties in finding labour for the harvests, finding transport facilities, finding service points for truck drivers, and finding shipping facilities for the international markets, it could maintain a nearly "business as usual" way of operation during the pandemic. Its environmental laws are among the strictest in the world, requiring sustainability standards from producers, which helped them to comply with restrictions and guidelines for mitigation while keeping up production, including regulations about the labour force, harvest practices and transportation measures [62].

Vietnam, another leading exporter, however, has suffered more decline. It has been considered one of the most vulnerable countries for virus spread, being geographically close to Wuhan, the centre of the pandemic. Vietnam, in spite of that, could combat the spread of the infection remarkably well. Although the pandemic has been contained, its impacts on Vietnamese society and the economy are apparent, including its coffee industry. Vietnam's coffee industry is largely export-based and is very sensitive to any change in the global market. The interrelatedness between the domestic and global markets means that the local coffee industry's success is only achievable in a stable global context. Thus, in spite of Vietnam's successful crops and success in containing COVID-19, the industry is strongly dependent on the global supply chain of coffee to be restored or reconnected. Local growers were poorly prepared to adapt to the new situation, while the local coffee industry has already been strongly impacted by an abnormally prolonged dry season that squeezed many of the farmers out economically. The outbreak of the pandemic has just plunged them even deeper into the crisis and forced the country to apply a lockdown and social distancing mechanism that resulted in bottlenecks in the flows of goods and services in the markets, including coffee. Vietnam's coffee industry has been seriously impacted by the pandemic when the country's access to its traditional import markets, such as the US, the UK, France and Japan, was almost blocked due to the disconnection of the global coffee supply chain, leaving only limited export possibilities to Spain and Germany. While the local coffee industry contributes greatly to Vietnam's economy, the industry lacks the necessary capacity to adapt to changes locally and globally, as the existing production and development model of the industry is merely productivity and market prone. That can promote rapid growth and development of the industry but does not prepare the industry for shocks or crises like COVID-19 [62].

Policy implications, therefore, can focus on the different roles countries play in the global network, identifying bottleneck and bridge countries that can play important intermediary roles in the governance of the global network. Higher densities in networks may be a sign of more interconnectedness and more diverse trade structures, and this may be advantageous in times of crises, adding more stability. The more central roles point to larger players, and they may attract more trading partners, especially in times of increasing uncertainties, which is visible in our results comparing network structures of 2018 and 2020. The formation of trading communities also supports this idea—the 2020 year of pandemic revealed the disappearance of trading communities with many small players, as these actors opted to join some larger groupings, and the surviving trading communities remained formed around major exporters. The only exception is the community formed around Belgium, a major intermediary with a geographically close network of other importer countries and one small exporter. The backbone of the network, reflected by the

MST, also unveils important features, eliciting the key linkages and groupings, showing central and peripheral actors. The change of this structure as a result of the pandemic may reflect the abilities of major players to cope with difficult situations and also the tendencies of less central actors to switch their partners. This backbone also reveals how shocks can be transmitted from region to region through the network. The network structure also unveils if countries form trade relationships based on geographic closeness and cultural and political similarities that allow a similar reaction to shock situations or if these considerations are neglected in contrast to other trade advantages. The MST structure shows how African producers are rather peripheral, as well as economically important countries such as China or Japan, while some European consumers can become central actors not only by their level of consumption but as intermediaries. The differing impacts of the pandemic show that even large producers may develop differently, as the comparison of Brazil and Vietnam reveals. This underlines the importance of a sustainability focus in contrast to a growth and productivity focus in the industry's development and the need for quality, added value and fairer trade conditions.

The present research, although producing a great deal of detailed information, suffers from some limitations and weaknesses. First, the network indicators measure the extent and direction of trade but cannot easily point to their causes. Our analysis attributed the changes found between the two years to the effects of the COVID-19 pandemic, but further analysis is needed to reveal the actual mechanisms leading to these results. This is a limitation of the present work, although it is reasonably assumed that the pandemic influenced international trade flows. Due to the lockdowns and travel and transport restrictions, long-distance trade was technically more complicated, if not impossible, production suffered due to limitations of labour force movement, and demand for foreign products also changed. The change in trade network parameters is probably due to these changes, although its extent is difficult to judge. Further research is planned to conduct a similar analysis for a longer time period, comparing the changes in 2020 to 'normal' years and to the 2008–2009 global financial crisis to see how the trade network parameters have changed in the past. More detailed work should be done regarding the regional patterns, too. With a longer time span, the 'normal' characteristics of the coffee trade network could be generalised, and then the changes attributable to the occurrence of crisis situations could be identified with more certainty. Further limitations of our work refer to data quality and the reliability of international trade values available in public databases. A continuation of the present work may be the assessment of the network dynamics for a longer time span, focusing on periods of economic and natural crises.

**Author Contributions:** Conceptualization, Z.B., M.F.-F. and M.I.M.; methodology, Z.B. and M.I.M.; software, Z.B. and M.I.M.; validation, Z.B., M.F.-F. and M.I.M.; formal analysis, Z.B. and M.I.M.; investigation, Z.B. and M.I.M.; resources, M.I.M.; data curation, M.I.M.; writing—original draft preparation, Z.B., M.F.-F. and M.I.M.; writing—review and editing, Z.B. and M.I.M.; visualization, M.I.M.; supervision, M.F.-F. and Z.B.; project administration, M.F.-F. and Z.B.; funding acquisition, M.F.-F. All authors have read and agreed to the published version of the manuscript.

**Funding:** This research received no external funding.

**Institutional Review Board Statement:** Not applicable.

**Informed Consent Statement:** Not applicable.

**Data Availability Statement:** Original data were downloaded from the publicly accessible database of the United Nations [33]. See https://comtradeplus.un.org for commodity 090111 (Coffee not roasted, not decaffeinated).

**Conflicts of Interest:** The authors declare no conflict of interest.

# Appendix A

**Table A1.** List of the 232 analysed areas (countries) with their three-digit codes.

| Label | Area | Label | Area | Label | Area | Label | Area | Label | Area | Label | Area | Label | Area |
|---|---|---|---|---|---|---|---|---|---|---|---|---|---|
| ABW | Aruba | CHL | Chile | GIB | Gibraltar | LBY | Libya | OMN | Oman | | | TCA | Turks & Caicos Isl' |
| AFG | Afghanistan | CHN | China | GIN | Guinea | LCA | Saint Lucia | PAK | Pakistan | | | TCD | Chad |
| AGO | Angola | CIV | Cote d'Ivoire | GMB | Gambia | LKA | Sri Lanka | PAN | Panama | | | TGO | Togo |
| AIA | Anguilla | CMR | Cameroon | GNB | Guinea-Bissau | LSO | Lesotho | PCN | Pitcairn | | | THA | Thailand |
| ALB | Albania | COD | CongoDemR | GNQ | Equatorial Guinea | LTU | Lithuania | PER | Peru | | | TJK | Tajikistan |
| AND | Andorra | COG | Congo | GRC | Greece | LUX | Luxembourg | PHL | Philippines | | | TKL | Tokelau |
| ANT | Netherlands Antilles | COK | Cook Islands | GRD | Grenada | LVA | Latvia | PLW | Palau | | | TKM | Turkmenistan |
| ARE | United Arab Emirates | COL | Colombia | GRL | Greenland | MAC | Macao-China | PNG | Papua New Guinea | | | TLS | Timor-Leste |
| ARG | Argentina | COM | Comoros | GTM | Guatemala | MAR | Morocco | POL | Poland | | | TON | Tonga |
| ARM | Armenia | CPV | Cabo Verde | GUM | Guam | MDA | Moldova-Rep | PRK | Korea-DemRep | | | TTO | Trinidad&Tobago |
| ASM | American Samoa | CRI | Costa Rica | GUY | Guyana | MDG | Madagascar | PRT | Portugal | | | TUN | Tunisia |
| ATF | French Southern& Antarctic Territories | CUB | Cuba | HKG | HongKong-China | MDV | Maldives | PRY | Paraguay | | | TUR | Türkiye |
| ATG | Antigua&Barbuda | CUW | Curacao | HND | Honduras | MEX | Mexico | PSE | Palestine | | | TUV | Tuvalu |
| AUS | Australia | CXR | Christmas Island | HRV | Croatia | MHL | Marshall Islands | PYF | French Polynesia | | | TWN | Taipei-Taiwan |
| AUT | Austria | CYM | Cayman Islands | HTI | Haiti | MKD | Macedonia-North | QAT | Qatar | | | TZA | Tanzania |
| AZE | Azerbaijan | CYP | Cyprus | HUN | Hungary | MLI | Mali | ROU | Romania | | | UGA | Uganda |
| BDI | Burundi | CZE | Czech Republic | ID | Countries | MLT | Malta | RUS | Russian Federation | | | UKR | Ukraine |
| BEL | Belgium | DEU | Germany | IDN | Indonesia | MMR | Myanmar | RWA | Rwanda | | | UMI | USA Minor Outlying Isl' |
| BEN | Benin | DJI | Djibouti | IND | India | MNE | Montenegro | SAU | Saudi Arabia | | | URY | Uruguay |
| BES | Bonaire-Sint Eustatius-Saba | DMA | Dominica | IO7 | British Indian Ocean Territory | MNG | Mongolia | SCG | Serbia & Montenegro | | | USA | United States of America |
| BFA | Burkina Faso | DNK | Denmark | IRL | Ireland | MNP | Northern Mariana Islands | SDN | Sudan | | | UZB | Uzbekistan |
| BGD | Bangladesh | DOM | Dominican Rep. | IRN | Iran-IslRep | MOZ | Mozambique | SEN | Senegal | | | VCT | Saint Vincent& the Grenadines |
| BGR | Bulgaria | DZA | Algeria | IRQ | Iraq | MRT | Mauritania | SGP | Singapore | | | VEN | Venezuela |
| BHR | Bahrain | ECU | Ecuador | ISL | Iceland | MSR | Montserrat | SHN | Saint Helena | | | VGB | British Virgin Islands |
| BHS | Bahamas | EGY | Egypt | ISR | Israel | MUS | Mauritius | SLB | Solomon Islands | | | VIR | British Antarctic Terr' |
| BIH | Bosnia&Herzegovina | ERI | Eritrea | ITA | Italy | MWI | Malawi | SLE | Sierra Leone | | | VNM | Vietnam |
| BLR | Belarus | ESH | Western Sahara | JAM | Jamaica | MYS | Malaysia | SLV | El Salvador | | | VUT | Vanuatu |

**Table A1.** *Cont.*

| Label | Area | Label | Area | Label | Area | Label | Area | Label | Area | Label | Area |
|---|---|---|---|---|---|---|---|---|---|---|---|
| BLZ | Belize | ESP | Spain | JOR | Jordan | MYT | Mayotte | SOM | Somalia | WLF | Wallis&Futuna Isl' |
| BMU | Bermuda | EST | Estonia | JPN | Japan | NAM | Namibia | SPM | Snt Pierre&Miquelon | WSM | Samoa |
| BOL | Bolivia | ETH | Ethiopia | KAZ | Kazakhstan | NCL | New Caledonia | SRB | Serbia | YEM | Yemen |
| BRA | Brazil | FIN | Finland | KEN | Kenya | NER | Niger | SSD | South Sudan | ZAF | South Africa |
| BRB | Barbados | FJI | Fiji | KGZ | Kyrgyzstan | NFK | Norfolk Island | STP | Sao Tome&Principe | ZMB | Zambia |
| BRN | Brunei Darussalam | FLK | Falkland Islands | KHM | Cambodia | NGA | Nigeria | SUR | Suriname | ZWE | Zimbabwe |
| BTN | Bhutan | FRA | France | KIR | Kiribati | NIC | Nicaragua | SVK | Slovakia | | |
| BVT | Bouvet Island | FRO | Faroe Islands | KNA | St Kitts&Nevis | NIU | Niue | SVN | Slovenia | | |
| BWA | Botswana | FSM | Micronesia | KOR | Korea-Republic | NLD | Netherlands | SWE | Sweden | | |
| CAF | Central African Rep. | GAB | Gabon | KWT | Kuwait | NOR | Norway | SWZ | Eswatini | | |
| CAN | Canada | GBR | United Kingdom | LAO | Lao DemRep | NPL | Nepal | SXM | St Maarten-Dutch | | |
| CCK | Cocos (Keeling) Isl' | GEO | Georgia | LBN | Lebanon | NRU | Nauru | SYC | Seychelles | | |
| CHE | Switzerland | GHA | Ghana | LBR | Liberia | NZL | New Zealand | SYR | Syrian Arab Rep. | | |

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
