# Peer review of "A Graph-Based Network Analysis of Global Coffee Trade—The Impact of COVID-19 on Trade Relations in 2020"

_sustainability, doi:10.3390/su15043289_

Round 1
Reviewer 1 Report
With regard to this manuscript, I have the following suggestions:
1. The picture of social network can be clearer. Can you delete unimportant labels to make the picture more intuitive.
2. The social network analysis in this manuscript only analyzes the international trade relations of coffee green beans, but the topic is the impact of COVID-19 on coffee green bean trade.
3. In the conclusion section, can you provide more research and analysis to support your conclusion, that is, why is the change in the trade structure of coffee green beans affected by COVID-19, and what are the possible impact mechanisms?
Author Response
Reviewer 1:
- The picture of social network can be clearer. Can you delete unimportant labels to make the picture more intuitive.
Reply: Figure 3a and 3b are changed. The label size is set proportional to number of trade partners, edge thickness is proportional to trade value. Only the countries with at least 20 trade partners, are shown. Similarly, the layout of Figure 5a and 5b are also adjusted.
- The social network analysis in this manuscript only analyzes the international trade relations of coffee green beans, but the topic is the impact of COVID-19 on coffee green bean trade.
Reply: Literature (more than 20 new sources) was added in the Introduction about the international trade theory, the international coffee trade, and about the impacts of Covid on international trade flows. The comparison of the pre-Covid year 2018 to the Covid-year2020 serves this purpose, although the difference between the two years cannot entirely be attributed to the pandemic. This issue is discussed as the limitation of the research at the end of Conclusions. Also,in page 7 we stated 4 research questions for the research. Then in the Materials and Methods section (page 13, Sub-Section 2.3) the theoretical aspects of analyzing them are explained.
- In the conclusion section, can you provide more research and analysis to support your conclusion, that is, why is the change in the trade structure of coffee green beans affected by COVID-19, and what are the possible impact mechanisms?
Reply: The Discussion and the Conclusion section is considerably extended, dealing with this issue, too.
Reviewer 2 Report
Dear authors,
In general, the paper is well structured and adequately referenced with all the cited references being relevant to the research. The methodological approach is appropriate for this type of analysis and the results are well presented. However, the theoretical and empirical background as well as the conclusions’ section need elaboration. More specifically:
Lines 20-22: Although you address the causality issue in the abstract, you don’t discuss this issue later in the main body of the paper. All this has to be discussed in the section of “discussion” or to add a section of limitations alternatively.
Introduction: The section of introduction is indefinite. You should explicitly declare which is the motivation to conduct the present study and additionally to “build” the conceptual framework of the study on this motivation.
Introduction: The literature review is limited on studies which have used similar methodologies. You should extend the literature review of the paper with studies which investigate the international trade of coffee particularly as well as with studies which investigate the differences in export patterns between pro-Covid and post Covid era.
Introduction: You should add a short text about the structure of the study at the end of introduction’s section.
Lines 224: You should indicate through which revision of HS classification you achieved the maximum data availability.
Lines 231-232: Did you considered the inflation effects between years 2018 and 2020? It is highly recommended to deflate the values by using a consumer price index in order to gain more “comparable” results.
Lines 232: Why only export values have been used? You should justify this. Generally, data for import flows is more trustworthy than data from export flows, as the exporter is motivated to report lower values of exported products for tax evasion.
Conclusions: The conclusions’ section needs elaboration. More specifically, you end up in a very general conclusion in which someone could come up just by studying the respective worldwide international trade figures in UNCOMTRADE database. Since you have available many results for analysis you should emphasize to the outcome of trade policy or to the direction of potential implications of these changes (relationships, positions) on trade patterns.
Author Response
Replies to Reviewer 2: Please see attachment.

Reviewer 3 Report
This study tries to identify the effect on the global trade network of coffee green 12 beans, comparing the Covid-year 2020 to the pre-Covid year 2018. I hope that my comments will be helpful for the authors to improve the atricle.
(1) I would like to suggest that the authors should explicitly explain the theoretical background behind the empirical model. At first, they should develop some theoretical model, and then they explain the empirical model based on the economic theory. Otherwise, this article becomes a simple exercise of statistical techniques.
(2) Results: The results need expansion — The results should be discussed and compared with other studies. Especially, the authors should clearly explain the similarity and differences between their results and those of previous research.
(3) Implications for research, practice and/or society: The paper must identify clearly any implications for research, practice and/or society. Thus, the paper must bridge the gap between theory and practice. How can the research be used in practice (economic and commercial impact), to influence public policy, in research? What is the impact on society? Are these implications consistent with the findings and conclusions of the paper?
(4) Relationship to Literature: Literature has not been extensively considered. The paper demonstrates an inadequate understanding of the relevant literature in the field and thus an appropriate range of literature sources and significant works are ignored.
Author Response
REVIEWER 3:
(1) I would like to suggest that the authors should explicitly explain the theoretical background behind the empirical model. At first, they should develop some theoretical model, and then they explain the empirical model based on the economic theory. Otherwise, this article becomes a simple exercise of statistical techniques.
Reply: In the second half of page 7 we stated four research questions, as motivations for the research. Then in the Materials and Methods section (on page 13, Sub-Section 2.3) these questions are addressed, and the theoretical aspects of handling and answering them are explained.
(2) Results: The results need expansion — The results should be discussed and compared with other studies. Especially, the authors should clearly explain the similarity and differences between their results and those of previous research.
Reply: The Discussion and the Conclusion section is considerably extended, dealing with the comparison to other studies and results, too.
(3) Implications for research, practice and/or society: The paper must identify clearly any implications for research, practice and/or society. Thus, the paper must bridge the gap between theory and practice. How can the research be used in practice (economic and commercial impact), to influence public policy, in research? What is the impact on society? Are these implications consistent with the findings and conclusions of the paper?
Reply: The Conclusions section is considerably extended, in which policy implications are discussed in detail.
(4) Relationship to Literature: Literature has not been extensively considered. The paper demonstrates an inadequate understanding of the relevant literature in the field and thus an appropriate range of literature sources and significant works are ignored
Reply: .Literature (more than 20 new sources) was added in the Introduction about the international trade theory, the international coffee trade, and about the impacts of Covid on international trade flows.
Many thanks for all your valuable comments and suggestions.
Round 2
Reviewer 2 Report
Dear authors,
I noticed extended modifications in your manuscript which unambiguously integrate the great majority of my comments. I hope that you find my suggestions helpful. To the best of my knowledge, all the detected issues have been addressed successfully.
Reviewer 3 Report
The article has been appropriately revised.